# DenseFusion-1M: Merging Vision Experts for Comprehensive Multimodal Perception

**Xiaotong Li**[1,2,][*] **Fan Zhang**[2][*], **Haiwen Diao**[3,2][*], **Yueze Wang**[2], **Xinlong Wang**[2][†], **Ling-Yu Duan**[1][†]

[1]Peking University [2]Beijing Academy of Artificial Intelligence (BAAI)

[3]Dalian University of Technology

Dataset: https://huggingface.co/datasets/BAAI/DenseFusion-1M

## Abstract

Existing Multimodal Large Language Models (MLLMs) increasingly emphasize complex understanding of various visual elements, including multiple objects, text information, and spatial relations. Their development for comprehensive visual perception hinges on the availability of high-quality image-text datasets that offer diverse visual elements and throughout image descriptions. However, the scarcity of such hyper-detailed datasets currently hinders progress within the MLLM community. The bottleneck stems from the limited perceptual capabilities of current caption engines, which fall short in providing complete and accurate annotations. To facilitate the cutting-edge research of MLLMs on comprehensive vision perception, we thereby propose *Perceptual Fusion*, using a low-budget but highly effective caption engine for complete and accurate image descriptions. Specifically, *Perceptual Fusion* integrates diverse perception experts as image priors to provide explicit information on visual elements and adopts an efficient MLLM as a centric pivot to mimic advanced MLLMs' perception abilities. We carefully select 1M highly representative images from uncurated LAION dataset and generate dense descriptions using our engine, dubbed DenseFusion-1M. Extensive experiments validate that our engine outperforms its counterparts, where the resulting dataset significantly improves the perception and cognition abilities of existing MLLMs across diverse vision-language benchmarks, especially with high-resolution images as inputs. The dataset and code are publicly available at https://github.com/baaivision/DenseFusion.

## 1 Introduction

Multimodal Large Language Models (MLLMs) [32, 12, 37, 3, 2, 39, 62, 9, 21, 44] have made remarkable strides in multi-modal understanding and reasoning by aligning the Large Vision Models (LVMs) [57, 46, 10] and Large Language Models (LLMs) [25, 54, 68]. To fully harness the capabilities of MLLMs in comprehensive visual perception, there is a critical demand for high-quality image-text datasets that provide dense and thorough descriptions across a wide range of visual elements. Such hyper-detailed datasets are essential for training MLLMs to accurately interpret and interact with diverse visual information. However, the scarcity of such rich datasets currently hampers the progress of the MLLM community. Given these challenges, it is crucial to pioneer a practical and efficient route to craft highly detailed image descriptions for comprehensive perception.

As the saying goes, "an image is worth a thousand words". Images contain various visual elements of different granularities that are essential yet challenging to harness. Employing human labor [43, 17] or advanced GPT-4V [43, 18, 8, 7] is one feasible option to generate accurate, reliable, and

---

[*]Equal contribution. † Correspondence to *lingyu@pku.edu.cn, wangxinlong@baai.ac.cn*.

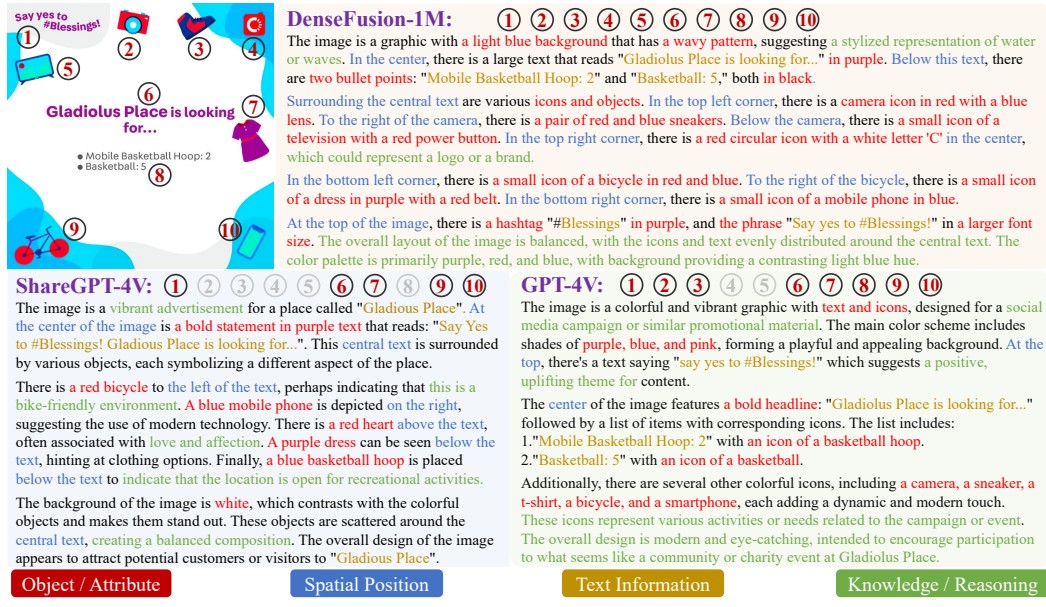

Figure 1: Illustration of the highly informative image description from DenseFusion-1M dataset and comparisons with state-of-the-art caption engine [8, 44]. It showcases comprehensive image understanding and captures all detailed visual clues (such as visual elements 1-10 in the image). For better visualization, information about objects/attributes, spatial positions, text information, and knowledge/reasoning are marked in individual colors.

high-quality image descriptions. Nevertheless, this approach is expensive and limits the scalability of the resulting dataset. Alternative strategies concentrate on caption engines [23, 32, 8], generating relatively detailed annotations over the web-crawled text. However, we observe that they often neglect many important visual details and still fall short of providing fine-grained descriptions with all visual clues. For example, the remarkable ShareGPT4V [8], struggles to accurately recognize various visual elements in Figure 1. The bottleneck lies in the limited perception capability of current caption engines for grasping diverse visual semantic information, including text recognition, object attributes, localization, and external knowledge, which hinders the sufficient exploration of visual information.

To address this issue, we empirically discover that incorporating diverse vision experts can effectively mitigate the limitations of caption engines' perceptual abilities. The perception information from specialized visual models can be considered as intermediate understanding of images. Typically, specialized perception models [42, 15, 22] outperform generalized MLLMs [65, 29, 22, 26, 70] within their respective visual specializations, *e.g.,* small object recognition for detection models. Therefore, utilizing these experts as strong assistants facilitates the perception process, enabling the efficient extraction of various visual elements for comprehensive image understanding. However, there remains less exploration into integrating their capabilities and diverse visual information to achieve well-rounded visual perception.

In this paper, we meticulously design a pipeline for comprehensive multimodal understanding, named *Perceptual Fusion*, integrating diverse vision experts as image priors and adopting a low-budget MLLM as a centric pivot for information fusion. Under this strategy, we exploit LAION [45], a valuable public resource, and delicately extract 1 million diverse and high-quality data. Firstly, we feed supplements from visual experts into the advanced GPT-4V and acquire 100K intricately detailed descriptions. With this meta dataset as guidance, we can efficiently develop a strong captioning engine capable of integrating strengths from multiple sources, including object detection, image tagging, text recognition experts for thoroughly comprehending image content. Leveraging this multimodal pivot, we can further construct a scalable, reliable, and high-quality pre-trained dataset, named DenseFusion-1M, enriched with abundant text information, accurate object and position recognition, and external knowledge. The hyper-detailed image-text data, in turn, enhances the perception of existing MLLMs to achieve better vision-language alignment.

In summary, our contributions are listed as follows:

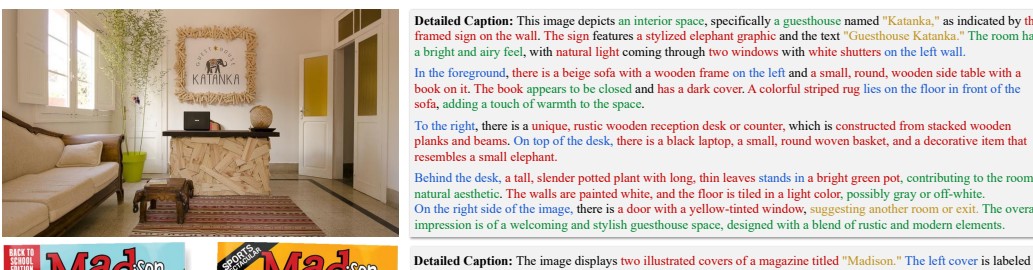
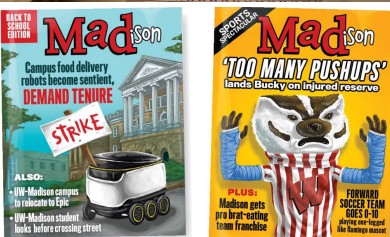
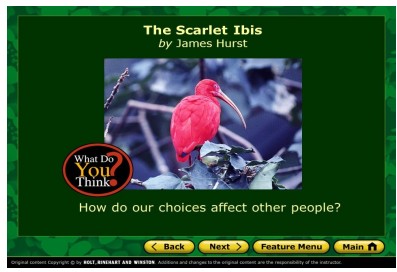

**Detailed Caption:** This image depicts an interior space, specifically a guesthouse named "Katanka," as indicated by the framed sign on the wall. The sign features a stylized elephant graphic and the text "Guesthouse Katanka." The room has a bright and airy feel, with natural light coming through two windows with white shutters on the left wall.

In the foreground, there is a beige sofa with a wooden frame on the left and a small, round, wooden side table with a book on it. The book appears to be closed and has a dark cover. A colorful striped rug lies on the floor in front of the sofa, adding a touch of warmth to the space.

To the right, there is a unique, rustic wooden reception desk or counter, which is constructed from stacked wooden planks and beams. On top of the desk, there is a black laptop, a small, round woven basket, and a decorative item that resembles a small elephant.

Behind the desk, a tall, slender potted plant with long, thin leaves stands in a bright green pot, contributing to the room's natural aesthetic. The walls are painted white, and the floor is tiled in a light color, possibly gray or off-white. On the right side of the image, there is a door with a yellow-tinted window, suggesting another room or exit. The overall impression is of a welcoming and stylish guesthouse space, designed with a blend of rustic and modern elements.

**Detailed Caption:** The image displays two illustrated covers of a magazine titled "Madison." The left cover is labeled as the "BACK TO SCHOOL EDITION" and features a cartoon depiction of a food delivery robot with a sign that reads "STRIKE" in front of a classical building resembling a campus structure. The text on this cover includes headlines such as "Campus food delivery robots become sentient, demand tenure," "UW-Madison campus to relocate to Epic," "UW-Madison student looks before crossing street," and "Also: UW-Madison student. "

The right cover is labeled as the "SPORTS SPECTACULARS EDITION" and features a cartoon of a character resembling Bucky Badger, a mascot associated with the University of Wisconsin, with a disgruntled expression. The headlines on this cover read "TOO MANY PUSHUPS lands Bucky on injured reserve", "Forward soccer team goes 0-10," and "Madison gets pro brat-eating team franchise."

Both covers have a colorful design with a mix of red, blue, yellow, and white colors. The background of each cover is different; the left cover has a blue sky with clouds, while the right cover has a yellow background. The text on both covers is in a mix of red and white colors, with the magazine title "Madison" prominently displayed in a large, bold red font. The overall layout of the image is symmetrical with the two covers side by side.

**Detailed Caption:** The image displays a vibrant red Scarlet Ibis perched on a green leafy branch. The bird's plumage is a uniform red, and its long, curved beak is also red. The Ibis is positioned slightly to the left of the center of the image, with its beak pointing to the right. The background is a blurred mix of green and brown hues, suggesting a natural, possibly forested environment.

In the foreground, there is a graphic overlay with a green background and text that reads "The Scarlet Ibis by James Hurst." Below this title, there is a circular logo with the text "What Do You Think?" in white on a red background. Beneath the logo, a question is posed in white text on a green background: "How do our choices affect other people?"

At the bottom of the image, there are navigation buttons with yellow text on a green background. From left to right, the buttons read "Back," "Next," "Feature Menu," and "Main." The bottom corner of the image contains a disclaimer in white text on a green background, stating: "Original content Copyright © by HOLT, RINEHART AND WINSTON. Additions and changes to the original content are the responsibility of the instructor."

The overall layout of the image suggests it is a digital educational resource, possibly a slide from a presentation or an interactive learning module. The inclusion of the Scarlet Ibis and the accompanying text implies a thematic connection to the short story "The Scarlet Ibis" by James Hurst, which is often used in educational settings to discuss themes of nature, disability, and the human spirit.

Figure 2: Examples of highly informative image descriptions from the DenseFusion-1M dataset, composed with various visual details and knowledge.

- To promote comprehensive visual perception, we introduce a perceptual fusion pipeline that leverages multi-source experts as image priors, establishing a low-budget yet powerful caption engine to comprehend image elements and generate well-crafted descriptions.

- Through our perception fusion strategy, we construct a large hyper-detailed image-text dataset, DenseFusion-1M with informative images and dense descriptions, including rich text information, multiple objects, attributes, spatial relations, world knowledge, etc.

- Based on our DenseFusion-1M, we validate that the trained MLLM demonstrates superior performance against existing state-of-the-art MLLMs across 10 vision-language benchmarks, especially for detailed text recognition and high-resolution image perception.

## 2 Related Work

**Large Multi-Modality Models:** The development of Large Multi-Modality Models (MLLMs) has witnessed significant advances in the abilities of comprehension and reasoning [32, 31, 12, 72, 39, 37, 9, 33, 13], typically through aligning pre-trained Large Vision Models (LVMs) [25, 54, 26, 68] with Large Language Models (LLMs) [10, 57, 46]. The pioneer works BLIP [32, 31], LLaVA [39, 37, 38], and Qwen series [39, 37] bridge the modality gaps through resamplers or MLP projectors, and obtain promising performances. Besides, Emu series [55, 53] exhibits strong in-context learning ability for multimodal content. Recently, there has been an emergent trend in developing high-resolution MLLMs [62, 38, 49, 21, 35, 63]. Among them, Monkey [35] resizes input images to fixed resolutions and divides them into multiple 448×448 patches, which are then processed by a pre-trained vision encoder. Moreover, CogAgent [21] utilizes low-resolution and high-resolution image encoders to recognize tiny visual elements inside a large image. LLaVA-NEXT [38], dubbed as LLaVA-1.6, introduces dynamic image aspect ratios and partitions the original images into multiple sub-images to capture more visual details, while LLaVA-UHD [62] divides images into smaller variable-sized slices for efficient and extensible encoding. Notably, Scaling on Scales $(S^2)$ [50] straightly extracts multi-scale features through image wrapping and rescaling without increasing image tokens. These high-resolution MLLMs capture tiny visual clues and benefit from meticulous image descriptions.

Hence, we aim to create hyper-detailed image annotations to enhance understanding of intricate visual elements and provide more accurate visual-language alignment.

**Image-Text Datasets:** Large-scale image-text datasets, e.g. LAION [45, 23], CC12M [6], Visual Genome [28] and YFCC [56], have effectively facilitate the development of vision-language pre-training. Along this line, BLIP-LAION [32] presents the synthetic short descriptions by the BLIP model, while LLaVA [39] and LLaVAR [69] prompt the text-only GPT-4 with visual information to generate conversations. Moreover, LaCLIP [14] rewrites the caption via ChatGPT through its in-context learning capability, while CapsFusion [66] leverages fine-tuned large language models to consolidate and refine information from both web-crawled and synthetic captions. To acquire detailed description datasets, recent studies seek help for the advanced GPT-4V model or human-in-the-loop strategy [8, 7, 60, 59, 18]. Among them, ShareGPT4V [8] comprises 100K captions from GPT-4V and employs an advanced captioner to produce an additional 1.2 million synthetic captions, while ALLaVA [7] directly leverages the advanced GPT4V's capabilities to create a synthetic dataset with detailed captions and instructional data. For region-level vision recognition, GLaMM [48] and all-seeing projects [60, 59] advance conversation generation with detailed region-level understanding and semantic tags. Lastly, ImageInWords (IIW) [17] presents 9k hyper-detailed captions through a human-in-the-loop annotation framework. DOCCI [43] instructs human annotators to create 19k comprehensive descriptions. Despite detailed visual annotations, their human-in-the-loop strategies require expensive labor costs and restrict the dataset scales. In contrast, we construct a low-budget caption engine empowered by diverse vision experts that can automatically generate large-scale and hyper-detailed image-text datasets at a negligible cost.

## 3   Methodology

In this section, we introduce the methodology design for constructing the dataset DenseFusion-1M. Specifically, we detail the data pre-processing pipeline for filtering high-quality image sources, the perceptual fusion procedure from vision experts, and the construction of the caption engine.

### 3.1   Data Processing

Establishing a high-quality dataset for comprehensive perception necessitates access to a large-scale data resource that encompasses a wide range of image categories and rich visual semantics.

Unlike methods such as ShareGPT4V[8], which meticulously curate images from specialized sources including COCO[36], SAM[27], Textcaps[51], etc, we opt to the widely-used LAION-2B [45] dataset, which naturally sources its diverse content directly from the wild internet, including different image categories like photos, posters, powerpoint, infographics, and more. Moreover, the LAION open-source dataset supports further academic research by offering readily accessible data that has been re-annotated by various studies [14, 8, 66, 23].

Despite its massive scale, LAION is still uncurated and contains significant issues about duplication [61], hindering both image diversity and quality. To address this, we mainly focus on two critical factors for data processing. Firstly, higher resolution images are prioritized since they generally provide richer visual content and more abundant semantics. Secondly, we emphasize the selection of representative images to preserve a greater diversity of visual content within the same data scale.

- **High Resolution Image Selection.** Images with a short-edge resolution less than 448 are filtered out to ensure the richness of the image content. Following this approach, approximately 500M images are retained from the initial 2B images, resulting in the subset named DenseFusion-500M.

- **Semantic Clustering and De-duplication.** To maximize the diversity of image distribution, we follow SemDeDup[1] to remove semantically duplicated images from DenseFusion-500M. Specifically, we employ k-means clustering on images features extracted via EVA-CLIP [54] to create 50,000 clusters. We set the threshold $\epsilon = 0.4$ to remove semantic duplicated images within each cluster, yielding an image set of 14 million images. Finally, we select the top 20 images from each cluster, which are closest to the cluster centers in the clustering process, to create our DenseFusion-1M from the deduplicated subset.

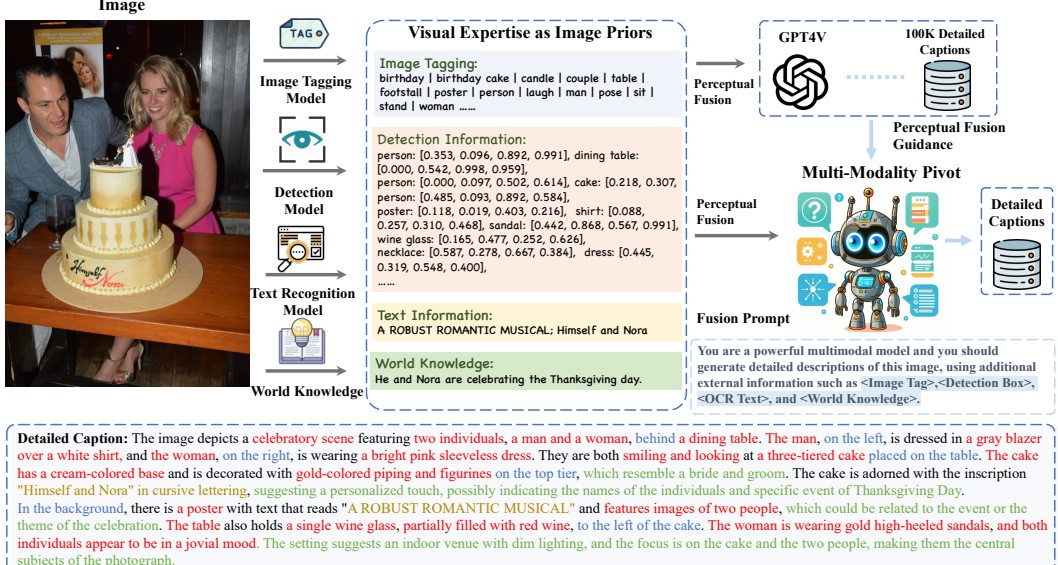

Figure 3: Pipeline of *Perceptual Fusion* to acquire DenseFusion-1M, which comprises 1 million hyper-detailed image descriptions. This pipeline leverages various visual experts as image priors and employs a multimodal model as the central pivot for integrating multi-source information. Its capability is learned from a 100K meta dataset generated by advanced GPT-4V.

## 3.2 Perceptual Fusion

Comprehensive visual perception is a prerequisite for multimodal understanding and reasoning. This perception ability can be achieved through extensive, detailed, and accurate alignments in image-text pre-training data. Despite the feasibility of current MLLMs [39, 32] for image captioning, they still struggle to provide meticulous descriptions. (1) Generalist MLLMs are designed for executing various instructions and are not intended for specific captioning tasks, especially for well-rounded image captioning. (2) Existing specialist caption engines lack a strong ability for comprehending and describing various visual elements inside high-resolution images, due to their inherent drawbacks in identifying all kinds of visual characteristics.

### 3.2.1 Mixture of Visual Experts

With the advancements in computer vision, numerous visual experts of various perceptual tasks have emerged and demonstrate outstanding capabilities within their respective domains [42, 15, 70, 22]. These models provide valuable intermediate perceptual information for image understanding. Therefore, comprehensively understanding the diverse visual elements in complex scenes can benefit from the collaboration of different specialists. In this section, we develop a perceptual fusion strategy with assistance from a variety of vision experts.

This approach specifically targets areas where generalist MLLMs often show limited perceptual capabilities. Our strategy includes the application of expert techniques in image tagging, object detection, text recognition, and the incorporation of world knowledge. We meticulously select these vision experts based on several key aspects of perception, which are detailed as follows.

- **Image Tagging**: Initially, we attempt to produce scene-level understanding for holistic images, including objects and visual scenes. Specifically, we employ the pre-trained RAM++ [70] that generates expansive tag descriptions over conventionally predefined tag categories. This approach enriches visual tag information and provides accurate scene annotations in overall image understanding, enhancing the recognition of diverse open-vocabulary concepts for object understanding.

- **Object Detection**: Comprehensive understanding relies on the perception ability of various object entities, while current MLLMs suffer from incomplete object perception and inaccurate positioning. Therefore, we utilize two types of specialized detection models to boost hyper-detailed recognition. (1) We employ the closed-set EVA02 [15] detection model trained on LVIS [20] and COCO [36]

to precisely detect the objects with basis concepts and varying sizes. (2) Meanwhile, we employ the open-set OWL-ViTv2 [42] detection model for capturing objects across broader categories constructed from the tagging classes. Afterward, we retain the objects with high confidence over the predefined threshold, and adopt a balanced sampling strategy to highlight small-scale objects, considering that the generalist MLLMs tend to focus on large-scale objects.

- **Text Recognition**: Text information is crucial for visual understanding, especially for documents, such as documents, posters, tables, and charts. However, generalist MLLMs often overlook some text elements and fail to identify text with various font styles and scales accurately. Meanwhile, we find that it is over 70% of the resulting images contain text information according to our statistics. Therefore, we employ OCR (Optical Character Recognition) models [22, 38] to recognize all textual elements within each image, even those with vague text information.

- **World Knowledge**: Although LAION's short captions crawled from the internet sometimes mis-align with image descriptions, they contain a wealth of world knowledge, including visual context, background information, and subtle details, etc. This can help boost the MLLMs' knowledge density and enhance the reasoning abilities. By incorporating these noisy yet rich captions, the models can achieve a deeper, more nuanced understanding of visual content, improving their performance in tasks requiring comprehensive visual and contextual understanding.

Here, we simultaneously integrate the image tags, objects, textual information, and external knowledge through the above vision experts [70, 15, 42, 22, 38, 45]. Through their powerful assistance, we facilitate the adaptive and meticulous perception capabilities of generalist MLLMs.

### 3.2.2 Perceptual Fusion Engine

To obtain precise and comprehensive image descriptions, the widely-used advanced GPT-4V [44] serves as an ideal MLLM with strong visual perception and contextual understanding capabilities. It can generate image descriptions that are further enriched with various visual information from specialized vision experts. Considering its expensive cost of time and finance, we attempt to construct an open-sourced and low-budget caption engine to efficiently mimic its ability for large-scale image captioning. We empirically discover that the perception ability of existing open-sourced caption engine can be enhanced with the assistance of additional visual experts, where they can improve the recognition of small-scale objects and OCR information, guiding our caption engine to focus on often overlooked content and correcting inaccuracies caused by its limited visual perception.

Initially, we adopt the proficient GPT-4V via manual-tuning prompts to generate image captions with extra visual information as the perceptual fusion guidance. The detailed prompt template can be found in Appendix. We thereby obtain 100K hyper-detailed image descriptions, i.e. DenseFusion-4V-100K. Using this meta dataset as guidance, we train our caption engine to learn from GPT-4V's characteristics and generate highly detailed image descriptions, as depicted in Figure 3. Our caption engine is based on LLaVA-1.6 (7B) [38], utilizing high-resolution images as inputs to ensure better visibility of detailed visual clues. The expertise of visual specialists are extracted offline and adopted as contextual information for caption engine. This process allows our engine to capture various visual clues effectively, enhancing its perception abilities by incorporating insights from vision experts. Consequently, it accurately identifies a wide range of objects and detailed textual information, resulting in image annotations with high information density.

### 3.3 Dataset Description

Utilizing the perceptual fusion pipeline, we incorporate insights from multiple visual experts into producing hyper-detailed image descriptions, resulting in the following datasets: (1) **DenseFusion-4V-100K.** GPT-4V generated 100K captions. (2) **DenseFusion-1M.** Scaling up to 1 million detailed captions by our caption engine. We conducted a statistical analysis to show the detailed dataset information in Table 1. On average, the captions are 190 words long and consist of 11 sentences with dense descriptions. As shown in the category distribution in Figure 4(b), the **DenseFusion** dataset contains diverse categories such as photos, visual art, commercial design, and infographics, making it a valuable resource with various image types. We employ LLaVA-1.5 [37] as a generalist MLLM for the category classification task. Generating hyper-detailed captions is fundamental to various multi-modal research tasks, as it facilitates the translation of images into language seamlessly. This capability presents significant potential in applications, *e.g.,* vision-language contrastive pre-training [25, 54], multimodal alignment in MLLMs [2, 39, 4], and text-conditioned image generation [47].

Table 1: Statistical information on DenseFusion-1M: (a) average number of characters, words, and sentences per caption, and (b) the lexical composition of the captions.

| Dataset Name | Caption | Samples | Char. | Word | Sen. | Nouns | Adj. | Adv. | Verb. | Num. |
|---|---|---|---|---|---|---|---|---|---|---|
| DenseFusion-4V-100K | GPT-4V | 100K | 1253 | 206 | 11.2 | 27.9% | 10.9% | 1.8% | 12.0% | 0.83% |
| DenseFusion-1M | Ours | 1059K | 1130 | 191 | 11.0 | 28.0% | 10.6% | 1.4% | 12.0% | 0.85% |

**(a) Diverse categories in DenseFusion dataset**   **(b) Category Distribution**

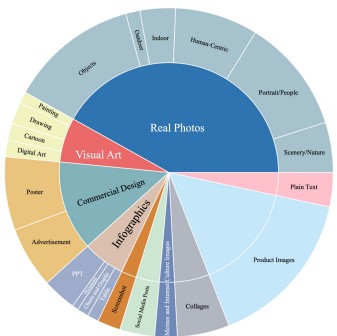

Figure 4: DenseFusion-1M dataset description. We obtain 1 million DenseFusion images after pre-processing. Figure (a) demonstrates the individual samples of classes and Figure (b) displays the category distribution, highlighting diverse images with rich semantics.

## 4 Experiments

In this section, we introduce the implementation details and compare the model trained by our DenseFusion-1M dataset with state-of-the-art MLLMs across diverse vision-language benchmarks. Finally, we validate the effectiveness of perception fusion qualitatively and quantitatively.

### 4.1 Implementation Details

**Caption Engine.** To explore the detailed visual clues inside each image, we adopt LLaVA-1.6 (7B) [38] to handle the high-resolution image inputs. For the meta dataset, we utilize GPT-4V to annotate the randomly selected 100K images from our picked 1M LAION data, thereby boosting our engine supported by various experts and producing high-quality annotations to mimic advanced GPT-4V. This supervised fine-tuning stage takes around ∼ 5.5 hours on 4 nodes of 8×A100 (40G) for 2 epochs. The visual knowledge from diverse visual experts are extracted and integrated as contextual information for the perception fusion prompt. Then we utilize the caption engine with the efficient deployment tool SGLang [71] to generate 1M data with enhanced multimodal perception.

**Evaluation Benchmarks.** To verify the efficacy of the provided DenseFusion-1M, we adopt these captions during the pre-training stage and follow the setup of LLaVA-1.5 [37] on various visual question answering (VQA) and multi-modality understanding benchmarks for evaluation, such as ScienceQA [41], TextQA [52], VQAv2 [19], GQA [24], SEED [30], MMBench [40], MME [16], POPE [34], MM-Vet [67], that covers a wide range dimensions for evaluating model abilities. The metric in Table 2 reflects the individual scores for each benchmark, typically represented as the percentage (%) of correct answers across all questions.

**Model Configuration.** To verify the effectiveness of DenseFusion-1M, we adopt it in the pre-training stage for vision-language alignment. The model is based on LLaVA-1.5 [37], using the vision encoder CLIP-ViT-L/14-336 [25] and the large language model (LLM) Vicuna [10] respectively. The vision encoder and LLM are connected by a two-layer multi-layer perception (MLP) projector. We utilize the approach of $S^2$ [49] for training the high-resolution MLLM, which is efficient in handling high-resolution inputs without increasing image tokens. We follow LLaVA-1.5 [37] that comprises a two-stage training stages. **(a) Pre-training Stage**. We first only train the projector for pre-alignment, then we conduct pre-training with a trainable vision encoder of the last 12 layers to further improve the perception ability. **(b) Instruction-tuning Stage.** For fair comparison, we follow LLaVA-1.5

Table 2: Comparisons with state-of-the-art approaches on 10 vision-language evaluation benchmarks, including SQA$^I$: ScienceQA-IMG [41], VQA$^{v2}$: VQA-v2[19], GQA [24], MME [16], POPE [34], VQA$^T$: TextVQA [52], MMB: MMBench [40], SEED [30], MM-Vet [67]. DenseFusion-1M is adopted in the pre-training stage for alignment, bringing significant and consistent improvements.

| Method | LLM | SQA$^I$ | VQA$^{v2}$ | GQA | VQA$^T$ | MME | MMB | SEED$^I$ | SEED | POPE | MM-Vet |
|---|---|---|---|---|---|---|---|---|---|---|---|
| *Low-resolution Multimodal Large Language Models* | | | | | | | | | | | |
| InstructBLIP | Vicuna-7B | 60.5 | - | 49.2 | 34.5 | - | 36.0 | - | 53.4 | - | 26.2 |
| QwenVL | Qwen-7B | 67.1 | 78.8 | - | 35.2 | - | 38.2 | - | - | 56.3 | - |
| QwenVL-Chat | Qwen-7B | 67.2 | 78.2 | 57.5 | 61.5 | 1487 | 60.6 | - | 58.2 | - | - |
| mPLUG-Owl2 | LLaMA2-7B | 68.7 | 79.4 | 56.1 | 58.2 | 1450 | 64.5 | - | 57.8 | - | 36.5 |
| InternVL-Chat | Vicuna-7B | - | 79.3 | 62.9 | 57.0 | 1525 | - | - | - | 86.4 | - |
| LVIS-4V | Vicuna-7B | 68.3 | 79.6 | 62.6 | 58.7 | 1528 | 66.2 | - | 60.6 | - | 31.5 |
| ShareGPT4V | Vicuna-7B | 68.4 | 80.6 | 63.3 | 60.4 | 1567 | 68.8 | 69.7 | 61.9 | 85.7 | 37.6 |
| LLaVA-1.5 | Vicuna-7B | 66.8 | 78.5 | 62.0 | 58.2 | 1510 | 64.3 | 66.2 | 58.6 | 85.9 | 30.5 |
| LLaVA-1.5 (Ours) | Vicuna-7B | 69.3 | 80.8 | 64.0 | 62.0 | 1574 | 69.2 | 70.1 | 62.3 | 86.5 | 37.8 |
| LLaVA-1.5 | LLaMA3-8B | 72.3 | 79.7 | 63.8 | 58.7 | 1553 | 72.8 | 69.2 | 61.8 | 85.0 | 34.9 |
| LLaVA-1.5 (Ours) | LLaMA3-8B | 72.9 | 80.4 | 64.4 | 61.0 | 1560 | 73.4 | 71.6 | 63.7 | 85.3 | 40.0 |
| LLaVA-1.5 | Qwen2-7B | 72.3 | 79.8 | 63.4 | 57.0 | 1566 | 72.9 | 70.0 | 62.5 | 85.7 | 35.8 |
| LLaVA-1.5 (Ours) | Qwen2-7B | 73.5 | 80.5 | 64.0 | 58.9 | 1528 | 73.5 | 71.6 | 63.6 | 86.0 | 41.4 |
| *High-resolution Multimodal Large Language Models* | | | | | | | | | | | |
| Moneky | Qwen-7B | 69.4 | 80.3 | 60.7 | - | - | - | - | - | - | - |
| LLaVA-1.6 | Vicuna-7B | 70.1 | 81.8 | 64.2 | 64.9 | 1519 | 67.4 | 70.2 | - | 86.5 | 43.9 |
| ShareGPT4V-S$^2$ | Vicuna-7B | 69.7 | 81.5 | 63.8 | 64.4 | 1547 | 68.0 | 70.1 | 62.4 | 86.7 | 35.0 |
| LLaVA-S$^2$ | Vicuna-7B | 68.2 | 79.7 | 63.3 | 60.8 | 1520 | 66.4 | 67.2 | 59.9 | 86.7 | 34.6 |
| LLaVA-S$^2$ (Ours) | Vicuna-7B | 72.1 | 81.6 | 65.3 | 67.4 | 1551 | 70.7 | 71.1 | 63.3 | 87.2 | 37.5 |

[37] and adopt the original LLaVA-mix-665K for instruction tuning, including GPT-generated and academic-oriented datasets. The detailed training recipe is shown in supplementary material.

## 4.2 Main Results

**Compared Models.** We report the experiment results against current state-of-the-art MLLMs, including Qwen-VL [4], InstructBLIP [11], mPLUG-Owl2 [64], InternVL [9], LLaVA-1.5 [37]. In particular, we compare our strategies with existing caption datasets or engines, e.g. ShareGPT4V [8], LVIS-4V [58]. To fully exploit its potential, we conduct comparisons under high-resolution settings with recent MLLMs, including Monkey [35], LLaVA-1.6 [38], and Scaling on Scales [49] (S$^2$).

**Experiment Results.** (1) Conventionally, Table 2 demonstrates that our meticulous descriptions significantly improve baseline models, providing solid and consistent benefits across all vision-language benchmarks, particularly for text-recognition scenes, e.g. TextVQA. Notably, our dataset originates from the generic LAION, which has no direct connection to the validation domains. Despite this, our strategies outperform ShareGPT4V, which uses images from COCO and VG that share a similar image distribution with the evaluation benchmarks, like VQAv2 and GQA. (2) Additionally, we observe that the potential benefits of our dataset are not fully exploited due to limited input resolutions, making MLLMs challenging to extract hyper-detailed image clues. To address this, we conduct further experiments using the high-resolution MLLM, Scaling on Scales [49] (S$^2$), which performs multi-scale aggregation on high-resolution inputs without increasing the number of image tokens. Even with a fifth of visual tokens of LLaVA-1.6 and requires no additional instruction tuning data, LLaVA-S$^2$ trained by our data achieves better performance than the state-of-the-art LLaVA-1.6 and exhibits higher forward efficiency. Besides, we reproduce LLaVA-S$^2$ using 1.2M pre-training data from ShareGPT4V [8], named ShareGPT4V-S$^2$, and we do not introduce additional supervised fine-tuning data for fair comparisons. Our dataset shows further gains compared to the low-resolution version, demonstrating our superiority in scenarios requiring hyper-detailed visual elements.

From the above results, we observe that (1) a high-quality image-text dataset is crucial during pre-training to enhance alignment across modalities before learning specific instruction patterns; (2) meticulous and accurate image descriptions are essential for high-resolution vision perception. Low-resolution MLLMs easily reach saturation due to blurred visuals and difficulty in exploring detailed clues. Therefore, meticulous image annotation is a promising direction for enhancing the hyper-detailed perception and reasoning capabilities of multimodal models.

## 4.3 Ablation Study

**Perceptual Fusion.** Generalist MLLMs occasionally exhibit inherent drawbacks in comprehensive perception, *e.g.,* omitting objects and weak in text recognition. For time saving, we performed

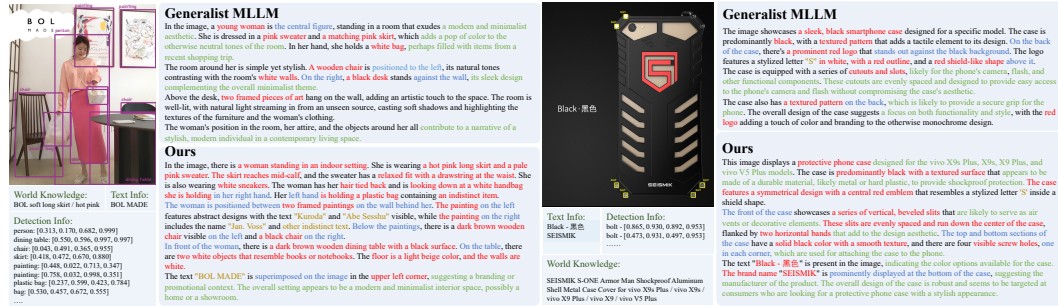

Figure 5: Visualization of perceptual fusion for enhanced image descriptions, which illustrates that the caption engine leverages additional information *e.g.,* text, object recognition, and world knowledge to produce detailed and comprehensive descriptions.

the ablation study using a subset of 100K data points from DenseFusion-1M as default setting. It is observed in Table 3, our strategy can effectively alleviate these issues, bringing substantial improvements on different benchmarks, especially in TextVQA with rich OCR information. We note that the relative improvement for high-resolution MLLMs becomes more emphasized, indicating that these MLLMs can benefit more from the visual details.

**Vision Encoder.** As demonstrated by previous studies [37, 8], unfreezing the vision encoder benefits from high-quality image-text alignment data. We verify the effectiveness of different training configurations: frozen vision encoder, half fine-tuning (last 12 layers), and full fine-tuning. Notably, fine-tuning improves performance, but full fine-tuning does not significantly outperform half fine-tuning. Therefore, we follow ShareGPT4V's approach of tuning the last 12 layers for fair comparisons.

Table 3: The effect of perceptual fusion with different resolution MLLMs.

| Model | MMB | SEED | $VQA^T$ | $SQA^I$ |
|---|---|---|---|---|
| Ours w/o fusion | 66.3 | 60.3 | 59.9 | 68.3 |
| Ours w fusion | 67.0 | 60.8 | 60.8 | 68.9 |
| Ours ($S^2$) w/o fusion | 66.9 | 60.8 | 61.7 | 68.7 |
| Ours ($S^2$) w fusion | 68.2 | 61.4 | 63.0 | 69.4 |

Table 4: Different number of trainable layers of vision encoders.

| Model | MMB | SEED | $VQA^T$ | $SQA^I$ |
|---|---|---|---|---|
| LLaVA-1.5 | 64.3 | 58.6 | 58.2 | 66.8 |
| Frozen | 65.7 | 59.8 | 59.6 | 68.8 |
| Half-tuning | 67.0 | 60.8 | 60.8 | 68.9 |
| Full-tuning | 67.3 | 60.9 | 61.0 | 67.2 |

**Visual Analysis.** We conduct the Visual analysis on specific contribution of visual experts for final description in Figure 6. Besides, We demonstrate caption examples from our perception fusion caption engine and the generalist MLLM LLaVA-1.6 7B [38] in Fig.5. Specially, the detected objects help the MLLM focus on individual objects, generating descriptions with more details and attributes. This integrated information allows the caption engine to achieve comprehensive image understanding for hyper-detailed captions. Note that even when not all additional information is provided, our caption engine can still focus on producing comprehensive captions, showcasing its robustness. More visualizations are included in the supplementary materials.

**Data Efficiency.** We conduct the experiment to verify the data efficiency of our high-quality image-text pairs across varying training samples. The experiment performances (%) demonstrate our superiority improvements than ShareGPT4V for equivalent data scale. This advantage becomes particularly significant with high-resolution inputs. The experiment indicates that the quality of detailed descriptions and input resolution significantly impact training effectiveness and hyper-detailed captions. As a result, the high-quality image-text data result in a more efficient training manner under the same data scale.

Table 5: Data efficiency under different number of training samples. (a) Low-resolution LLaVA-1.5. (b) High-resolution LLaVA-$S^2$.

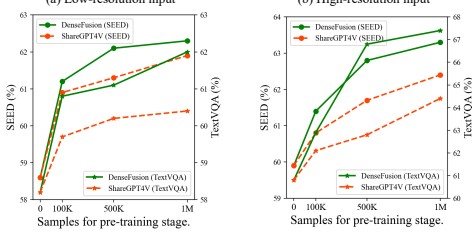

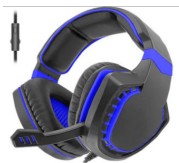

**Detailed Caption**
*(with World Knowledge):*
The image displays a **Sentry Gaming Headset, model HPX-GX250B.** The headset features a predominantly black color scheme with striking blue accents ……

**Detailed Caption**
*(without World Knowledge):*
The image displays a pair of over-ear headphones with a predominantly black color scheme accented with vibrant blue highlights ……

**World Knowledge:** (Product Information)
**Sentry Gaming Headset, HPX-GX250Bv**

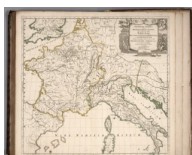

**Detailed Caption**
*(with World Knowledge):*
This image displays an aged map **with a focus on Europe, including parts of France, Germany, Italy, and the Mediterranean region.** The map is detailed, showing geographical features such as rivers, mountain ranges, and coastal outlines. ……

**Detailed Caption**
*(without World Knowledge):*
This image displays an aged map with boarders. The map is detailed, showing various countries, bodies of water, and geographical features…….

**World Knowledge:**
(Background)
**Europe, France, Germany, Italy and Mediterranean Map**

(a) Visual analysis on specific contribution of **World Knowledge** for final description, enhancing the knowledge depth beyond perceptual models and increasing its information density.

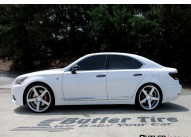

**Detailed Caption**
*(with Text Information):*
……, On the concrete surface, directly in front of the car, there is a large, stylized text that reads **"Butler Tire"** followed by the slogan **"We Baby Your Car."** Below the slogan, there is a website address **"butlertire.com"** and a phone number **"770.423.8473."** The text is in a bold, sans-serif font, and the color of the text contrasts with the concrete, making it prominent, ……

**Text Information (OCR)**
**Butler Tire;**
**We Baby Your Car;**
**butlertire.com;**
**770.423.8473.**

**Detailed Caption**
*(without Text Information):*
……, On the concrete surface, directly in front of the car, there is a large text logo that reads **"Butler Tire We Baby Your Car"** in stylized lettering. The logo is in black with a red underline, and it spans across the width of the image. The text is clear and legible, indicating the name of a business that specializes in car care or tire services, ……

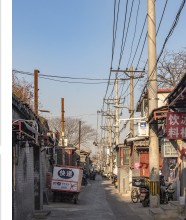

**Detailed Caption**
*(with Text Information):*
……, On the left side of the alley, there is a cart with a sign that reads **"快递"** and **"圆通速递."** which are Chinese words for express delivery and fast delivery. The cart is red and white, with additional text that includes **"车辆编码：003-13637,"** likely a vehicle code for delivery purposes, The right side of the alley shows more commercial activity, with a storefront displaying Chinese characters and a sign that includes the word **"饮料冰柜."** which translates to "beverage shaker.",……

**Text Information(OCR)**
快递:圆通速递； 饮料冰柜
车辆编码： 003-13637

**Detailed Caption**
*(without Text Information):*
……, On the left side of the alley, there is a red cart with a white sign **Chinese characters** and the logo of a company that appears to be related to automotive services Some of the buildings have **signage, including one with Chinese characters,** suggesting a locale where the language is prevalent, ……

(b) Visual analysis on specific contribution of **Text Recognition (OCR)** model for final description, providing accurate text information with different data types.

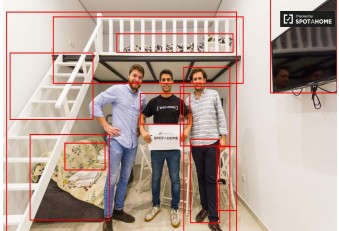

**Detection Information**
**bunk bed: [0.01, 0.07, 0.70, 0.99], stair: [0.01, 0.10, 0.30, 1.00]**
**bunk_bed: [0.15, 0.09, 0.72, 0.37], person: [0.26, 0.28, 0.43, 0.99]**
**person: [0.40, 0.31, 0.57, 0.97], person: [0.54, 0.30, 0.68, 0.99]**
**couch: [0.08, 0.58, 0.32, 0.94], tv: [0.80, 0.00, 1.00, 0.41]**
**stair: [0.02, 0.24, 0.27, 0.52], shirt: [0.26, 0.36, 0.43, 0.64]**
**shirt: [0.53, 0.38, 0.68, 0.63], trousers: [0.56, 0.60, 0.65, 0.95]**
**chair: [0.47, 0.66, 0.57, 0.92], bed: [0.34, 0.16, 0.69, 0.24]**
**jersey: [0.41, 0.41, 0.56, 0.54], chair: [0.64, 0.63, 0.70, 0.90]**
**······**

**Detailed Caption**
*(with Detection Information):*
The image displays an interior space with three men standing in the center. **The man** on the left is wearing a **light blue shirt and jeans**, with his hands in his pockets. The man in the middle is holding a sign with the text "SPOTAHOME**"** and is dressed in a **black t-shirt** with the same logo, **blue jeans, and sneakers.** The man on the right is wearing a **striped shirt with horizontal stripes in shades of blue and white, dark trousers, and brown shoes.**

Behind the men is a **white bunk bed with black railings**, and the **lower bunk** is covered with a **patterned blanket and pillows.** To the left of **the men is a white staircase** with a simple design leading up to **the bunk bed.** On the right side of the image, there is a **flat-screen TV mounted on the wall**, and below it, there is **a white chair.**

In the foreground, there is **a small couch with a floral pattern**, and **a folded blanket** is placed on it. The **floor is a light wood**, and the walls are white, creating a bright and minimalistic atmosphere. The room appears to be a showroom or a staged living space, as suggested by the cleanliness and the presence of the sign with the brand name.

**Detailed Caption**
*(without Detection Information):* *missed: couch, blanket, chairs; wrong: table, cushion;*

Three men stand in an interior space with a modern and minimalistic design. **The man** on the left wears a **light blue shirt and jeans**, with his hands in his pockets. The man in the center holds a sign with the text "SPOTAHOME" and wears a **black T-shirt** with the same text, along with dark pants and sneakers. The man on the right is dressed in a patterned shirt with vertical stripes, **dark pants, and brown shoes.**

Behind them is a **white bunk bed with a black mattress** on the upper level, and a **floral-patterned blanket** is visible on the lower bunk. To the right of the men is a **flat-screen television** mounted on the wall. The floor is a light wood, and the walls are white, contributing to the room's bright and airy feel.

The room also features a **white staircase** with a simple design leading to the upper bunk. On the left side of the image, there is a small ~~white table~~ with a floral-patterned cloth on it, and **a white cushion** is placed on the floor nearby. The overall atmosphere of the room is clean and organized, with a focus on functionality and a limited color palette dominated by white and natural wood tones.

(c) Visual analysis on specific contribution of **Object Detection** model for final description, effectively capturing various visual objects of different granularities and enhancing the precision as well as recall.

Figure 6: Visualization on specific contribution of world knowledge, text recognition, and detection model to produce detailed and comprehensive descriptions.

# 5 Conclusion

In this paper, we tackle the challenge of limited high-quality image-text data by developing a low-budget caption engine for high-resolution images and hyper-detailed captions. Our strategy involves curating a dataset from the LAION-2B corpus, followed by a perceptual fusion pipeline that guides a multimodal model to integrate information from various vision experts and thereby yields one million well-rounded descriptions, dubbed DenseFusion-1M. We believe that such an extensive image-text dataset, characterized by its hyper-detailed nature, would substantially enhance the capabilities of MLLMs by enabling more effective alignment between visual and textual data.

# 6  Acknowledgement

This work was supported by the Program of Beijing Municipal Science and Technology Commission Foundation (No.Z241100003524010), in part by the National Natural Science Foundation of China under Grant 62088102 and the National Key R&D Program of China (2022ZD0116302), in part by AI Joint Lab of Future Urban Infrastructure sponsored by Fuzhou Chengtou New Infrastructure Group and Boyun Vision Co. Ltd, and in part by the PKU-NTU Joint Research Institute (JRI) sponsored by a donation from the Ng Teng Fong Charitable Foundation.

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

# A Overview

Dataset of **DenseFusion-1M** will be open sourced at https://huggingface.co/datasets/BAAI/DenseFusion-1M. In this Appendix, we present brief description of our dataset in Sec. B. Sec. C presents implementation details of our framework. Besides, more examples and results are visualized in Sec. F.

# B Dataset

The dataset, named **DenseFusion-1M**, is a large-scale image description dataset designed to enhance the perceptual abilities of Multimodal Large Language Models (MLLMs). It contains 1 million hyper-detailed image descriptions derived from a subset of the LAION dataset, carefully curated and annotated using our caption engine with *perceptual fusion* that integrates diverse vision experts.

# C Implementation

## C.1 Training Details.

The main training implementations are outlined in the primary paper. In this section, we detail the hyper-parameters used to train the MLLM for evaluating our data. During the pre-alignment stage, we exclusively train the projector, resulting in more stable and slightly enhanced performance. In the pre-training phase, we unfreeze the Vision Encoder (VE) for the last 12 layers, the Language Model (LM), and the projector. For instruction tuning, we utilize the original data from LLaVA-1.5 and the LLaVA-mix-665K instruction tuning dataset to fine-tune both the projector and language model.

Table 6: Hyper-parameter settings for the training details.

| Hyperparameter | Pre-aligning | Pre-training | Instruction Tuning |
|---|---|---|---|
| Batch Size | 256 | 256 | 128 |
| Learning Rate (lr) | 2e-5 | 2e-5 | 2e-5 |
| LR Schedule | | cosine decay | |
| LR Warmup Ratio | 0.01 | 0.01 | 0.01 |
| Weight Decay | 0 | 0 | 0 |
| Trainable Module | Projector | Projector,VE,LM | Projector,LM |
| Epoch | 1 | 1 | 1 |
| Optimizer | | AdamW | |
| DeepSpeed stage | 3 | 3 | 3 |

## C.2 Prompt Engineering

The constructing pipeline leverages prompt engineering to generate hyper-detailed image descriptions. This process involves carefully crafting prompts that guide the advanced GPT-4V to produce comprehensive and accurate annotations. The prompts are designed to integrate insights from various vision experts, enhancing the overall quality and granularity of the dataset.

**Prompt for GPT-4V.** We use the following prompt to guide GPT-4V to generate the detailed caption of given images.

```
You are the most powerful large multimodal model which is responsible for generating image description to help the blind
    people to understand the world. Since they cannot see, so you should describe the image as detailed as possible.

The description of image must abide by the following policies:
    1. The generated caption must be comprehensive and detailed plain text, covering as many aspects / content / areas /
       contents of the image as possible.
    2. You may describe the foreground / background / salient objects.
    3. When describing objects, please endeavor to include as much of the following information:
       3.1. textures / attributes / locations / presence / status / characteristics / numbers of objects
       3.2. relative positions between objects
    4. The composition / color / layout / texture of image should also be considered.
    5. You may describe the elements one by one with details.
    6. If there are common sense or world knowledge, for example, species, celebrities, scenic spots and historical sites,
       you must state them explicitly instead of using phrases like "a person", "a place", etc.
    7. Other objective and subjective details that can help understand and reproduce the image.
    8. Text contents must be appeared in the caption if there exists. Keep the original language of text content.
```

```
 9. The description should be purely factual, with no subjective speculation.
10. If there are some statement are inferred, just state the conclusion. DO NOT add the evidence or thought chain.
11. DO NOT add description associated with aspects like mood or atmosphere.
12. DO NOT including any reasoning description like "probably because" or "appears to be"
13. DO NOT add any unnecessary speculation about the things that are not part of the image such as "the image is
      inspiring to viewers" or "seeing this makes you feel joy".
14. DO NOT add things such as "creates a unique and entertaining visual", as these descriptions are interpretations and
      not a part of the image itself.
15. DO NOT analyze the text content in the image, and only tell the content themselves.
16. DO NOT add any further analysis to the image.
17. DO NOT use introductory phrases like "The image showcases", "The photo captures", "The image shows" and more.
18. The caption should NO longer than 192 words.
Besides image, you are also provided with some external information to help you understanding the image including a short
      caption, detection results, ocr results, attributes, etc. The short caption might contains rich world knowledge which
      should be considered in the final caption but also may not have any relevance to the image. Besides, there might be
      some errors in the external information including detail missing or wrong details. If there are mistakes, you may
      ignore them. Note that external information like bounding box are just a reference information, some details like
      bounding box should not be presented in the final caption since it's not a common information in caption. If the
      external information is not used, DO NOT specify the reason of not using them.
[External Information]:
    [World Knowledge]: {SHORT CAPTION}
    [Detection Box]:
        {OBJECT AA}: [x1, y1, x2, y2]
        {OBJECT BB}: [x1, y1, x2, y2]
        ...
    [OCR]:
        {SENTENCE A}
        {SENTENCE B}
        ...

[IMAGE]:
```

**Prompt for Caption Engine.** We use the following prompt to prompt our caption engine to generate the detailed caption of given images. Due to the supervision from the meta-dataset of GPT-4V, this prompt can be designed rather simple.

```
You are a powerful multimodal model and you should generate detailed descriptions of this image, using additional external
      information such as [Caption], [Detection Box], and [OCR]. [Caption] might contain rich world knowledge which should
      be considered in the final description but also may not have any relevance to the image. Although this information may
      contain errors or be incomplete, you should disregard any inaccuracies. External details like detection boxes are
      just for reference and should not be included in the final description. If external information is not used, do not
      specify why.
[External Information]:
    [World Knowledge]: {SHORT CAPTION}
    [Detection Box]:
        {OBJECT AA}: [x1, y1, x2, y2]
        {OBJECT BB}: [x1, y1, x2, y2]
        ...
    [OCR]:
        {SENTENCE A}
        {SENTENCE B}
        ...
[IMAGE]:
```

# D   Approaches of data cleaning

## D.1   Data Processing and Bias Mitigation

Images from LAION are processed using automated filtering techniques, such as CLIP embeddings and specialized classifiers, to remove harmful or inappropriate content. To address potential biases, we employ a Semantic Clustering strategy during data pre-processing, which involves balanced sampling based on image semantics to promote diversity and mitigate biases from uneven distribution. We plan to enhance our approach by implementing additional strategies, including diverse source construction, where we will extend our dataset by integrating images from multiple sources to improve data diversity and balance.

In the future work, we aim to develop a Large Language Model (LLM) that evaluates image and caption content to systematically address any observed biases. For pre-processing visual perceptions, we prioritize advanced visual experts in specific domains to ensure high-quality and reliable predictions, implement a high-confidence threshold for visual expert predictions to effectively manage noise, and use a carefully designed prompt template for GPT-4V to refine inaccuracies based on real image content. In ensuring quality in hyper-detailed descriptions, we iteratively refine our pipeline to construct a high-quality dataset, review it to eliminate low-quality captions through heuristic rules, and plan to develop an LLM to evaluate descriptions and systematically filter out various types of bias.

### D.2 Quality Control

In our pre-processing of visual perceptions, we prioritize advanced visual experts in specific domains to ensure high-quality and reliable predictions. To effectively manage noise, we implement a high-confidence threshold for visual experts' predictions, as detailed in Section 3.2.1. Additionally, we have carefully designed a prompt template for GPT-4V that refines inaccuracies from visual experts based on real image content, thereby encouraging our engine to mitigate noise. Regarding the quality of hyper-detailed descriptions, we iteratively refine our pipeline in the early stages to optimize high-quality dataset construction. Following this, we review the entire dataset and eliminate low-quality captions using heuristic rules to address issues such as repetition and incompleteness. Furthermore, we plan to develop a Large Language Model (LLM) to evaluate the descriptions and systematically filter out various types of bias.

## E   Limitation and Discussion

For comprehensive visual perception, we propose *perceptual fusion*, efficiently integrating the insights from visual experts and constructing the DenseFusion-1M dataset with well-rounded descriptions. Despite its promising results, some issues could be improved: (1) Limited by the current capacity of MLLMs, describing all visible information in an over-complicated image perfectly is still hard to be ensured. (2) The information gathered by visual experts is inevitably noisy; therefore, only high-confidence feedback is retained to ensure accuracy. This can be enhanced by incorporating more sophisticated vision experts. (3) Given the caption engine's contextual capabilities, only the most crucial visual base models are selected to ensure the full exploitation of contextual information. As the language model's contextual understanding advances, additional details, such as dense region comprehension, can be progressively integrated.

## F   Visualizations on DenseFusion-1M

**Detailed Caption.** We provide more examples of image captions in Tab. 7 and Tab. 8. Besides, to further evaluate the consistency between original images and generated captions, we use Dall-E 3 [5] to reconstruct the images based on the generated captions. The comparative results from different caption engines are illustrated in Fig. 7. Compared to other caption engines, our model demonstrates significant advancements in terms of element consistency, spatial relationships, and accuracy. This also indicates the potential of our datasets for conditional image generation tasks which we leave it for future research.

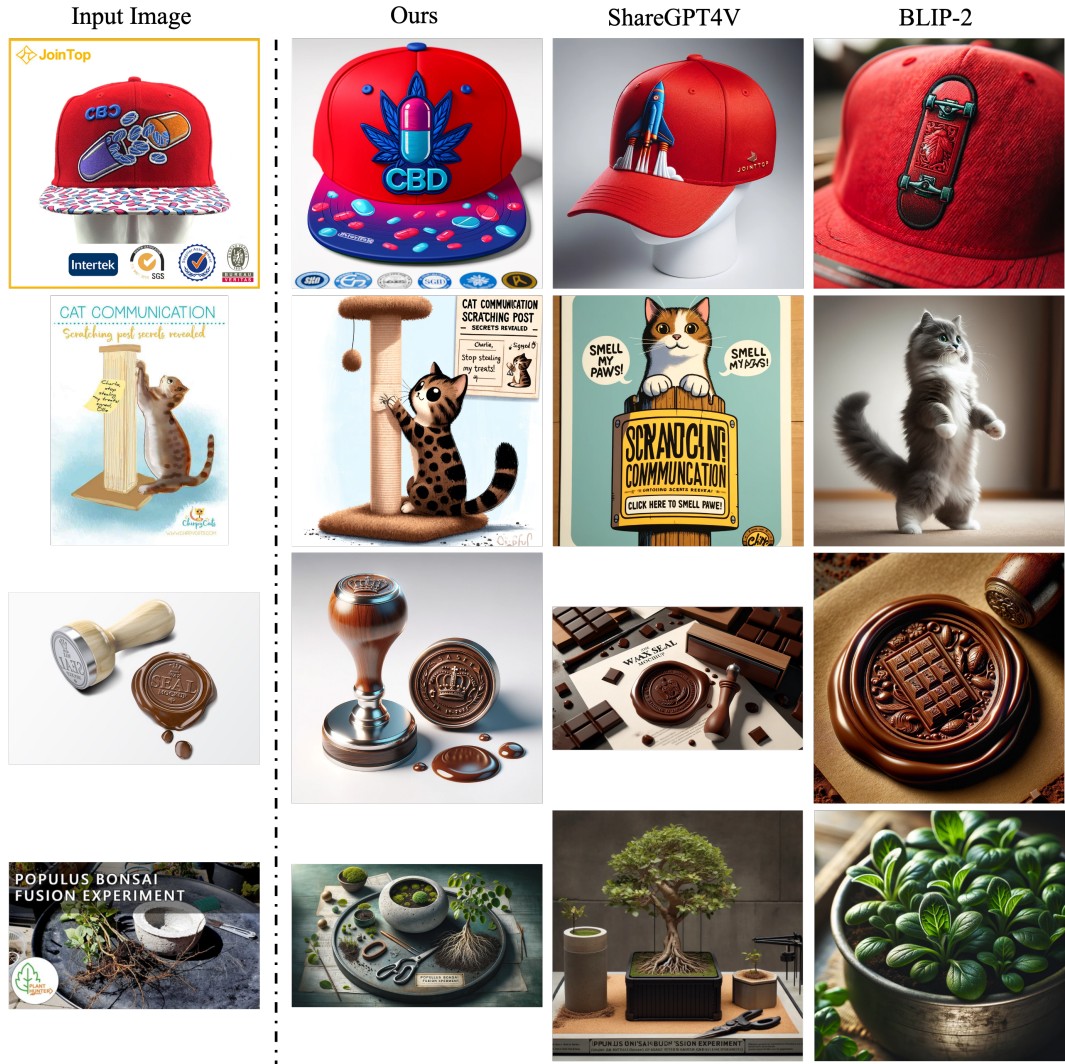

Figure 7: We use DALL-E 3[5] to reconstruct images from captions generated by our model, ShareGPT4V[8], and BLIP-2[31] to assess caption-image consistency. The reconstructed images using BLIP-2 differ significantly from the original ones, indicating that the captions lack substantial information or detail. Meanwhile, ShareGPT4V's reconstructions often include incorrect or absent elements, indicating its limited perceptual capabilities. Compared to ShareGPT4V and BLIP-2, our method demonstrates greater consistency between the reconstructed images and the original ones, showing the potential for controllable image editing.

**Visualizations on the image descriptions.**

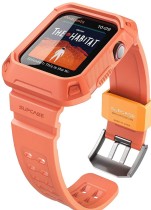

**Detailed Caption:** This image displays a SUPCASE Unicorn Beetle Pro protective case designed for the Apple Watch Series 6/SE/5/4 with a 44mm size. The case is presented in a two-part view: the top part shows the case attached to an Apple Watch, and the bottom part shows the case without the watch. The case is a rich, coral-orange color with a rugged texture, featuring a mix of matte and slightly glossy finishes. It has a rectangular shape with rounded corners and a prominent bezel that appears to be reinforced for added protection. The case has a built-in strap band, which is also coral-orange, with a smooth texture and a matte finish. On the top part of the case, there is a digital display showing the time as "10:09" and the podcast "The Habitat" with a visual of a person standing on a grid. Below the display, the word "Episode: I This Is the Way" is visible. The digital display is encased within the protective case. The bottom part of the case has a cut-out section that reveals the Apple Watch's digital crown and side button, as well as the charging port. The cut-outs are precisely designed to accommodate these features without obstructing them. The SUPCASE logo is visible on the side of the case, and the website "www.SUPCASE.com" is printed on the strap band. The text "MADE IN CHINA" is also present on the strap band, indicating the manufacturing origin of the product. The overall layout of the image is clean, with the product centered against a white background, emphasizing the case and its features. The image is well-lit, with no shadows obscuring the details of the case.

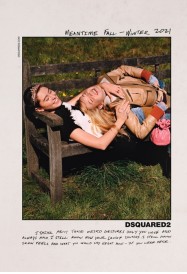

**Detailed Caption:** The image displays two individuals on a wooden bench with a grassy background. The person on the left appears to be wearing a black and white patterned top, a tiara, and has dark hair. The individual on the right is dressed in a beige jacket with red cuffs, a white shirt, and is wearing multiple rings on their fingers. Both are smiling and looking upwards, seemingly engaged in a joyful interaction. The bench is positioned on grass, and the background is a blurred natural setting, suggesting an outdoor environment. The image also contains text. At the top, the text reads "Meantime Fall - Winter 2021." Below the individuals, additional text states "think about these weird gestures only you have and always had. I still know how your laugh sounds. I still know how your skin feels, and what you would say now - if you were here." At the bottom of the image, the text "DSQUARED2" is prominently displayed. The overall color palette of the image is warm, with the natural tones of the wood and grass contrasting with the clothing of the individuals.

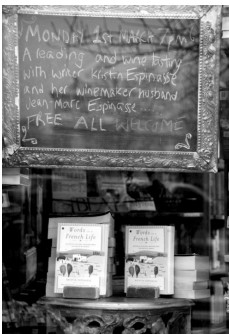

**Detailed Caption:** The image is a black and white photograph featuring the interior of a bookstore. In the foreground, there is a display table with several copies of a book titled "Words French Life" by Kristin Espinasse, prominently placed in the center. The book covers show illustrations of French countryside scenes. Behind the table, a large blackboard spans the width of the image, with white chalk text that reads: "Monday 1st March 7pm A reading and wine tasting with writer Kristin Espinasse and her winemaker husband Jean Marc Espinasse FREE ALL WELCOME." The text is written in a casual, handwritten style. The blackboard is framed by an ornate, decorative silver frame, which adds a touch of elegance to the setting. The reflection of the surrounding environment can be seen on the glass surface of the blackboard, indicating that the photograph was taken from the outside looking in. The bookstore's shelves are filled with various books, though the specific titles are not clearly visible. The overall composition of the image, with its focus on the book and the announcement of the event, suggests that this photograph was taken to promote the reading and wine tasting event mentioned on the blackboard.

Table 7: Visualizations on the image description in **DenseFusion-1M**. For better visualization, information about objects/attributes, spatial positions, text information, and knowledge/reasoning are marked in individual colors.

**Visualizations on the image descriptions.**

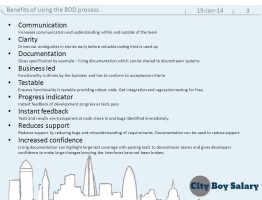

**Detailed Caption:** The image displays a slide with a title and bullet points, detailing the benefits of using the Behavior-Driven Development (BDD) process. The title at the top reads "Benefits of using the BDD process" with a date "19-Jan-14" and a page number "3" aligned to the right. The bullet points listed are as follows:
- Communication: Increases communication and understanding within and outside of the team.
- Clarity: Drives out ambiguities in stories early before valuable coding time is used up.
- Documentation: Gives specification by example – living documentation which can be shared to downstream systems.
- Business led: Functionality is driven by the business and has to conform to acceptance criteria.
- Testable: Ensures functionality is testable providing robust code.
- Progress indicator: Instant feedback of development progress as tests pass.
- Reduces support: Reduces support by reducing bugs and misunderstanding of requirements. Documentation can be used to reduce support.
- Increased confidence: Living documentation can highlight large test coverage with passing tests to downstream teams and gives developers confidence to make large changes knowing the interfaces have not been broken.
The slide background is white, and the text is predominantly black with the title in blue. The bottom of the slide features a stylized graphic of a city skyline with notable buildings, such as the Empire State Building and the Leaning Tower of Pisa, in a simplified black outline. The skyline is set against a light blue background. In the lower right corner, there is a logo with the text "City Boy Salary" in a darker blue, and the name "Simon Powers" in a smaller font, followed by the year "2014." The overall layout is professional and appears to be part of a presentation or educational material.

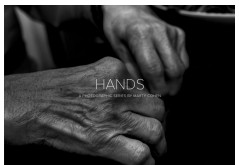

**Detailed Caption:** The image is a black and white photograph focusing on the detailed textures and lines of human hands. In the foreground, a hand with visible veins and creases is clenched into a fist, with the thumb tucked in and the other fingers extended outward. The hand appears to be that of an older individual, given the prominent veins and the aged skin texture. The background is out of focus, with the suggestion of another hand and possibly a bowl or a rounded object, but these details are indistinct due to the shallow depth of field. The composition of the image places the hand in the lower left quadrant, drawing the viewer's attention to the intricate details of the skin and the muscle structure of the hand. The lighting is soft and diffused, creating subtle shadows and highlights that accentuate the contours and lines of the hand. Overlaying the image is text that reads "HANDS" in large, white, capitalized letters, followed by a smaller subtitle "A PHOTOGRAPHIC SERIES BY MARTY COHEN." The text is clear and contrasts with the darker tones of the photograph, ensuring readability. The font is modern and sans-serif, which complements the contemporary feel of the photographic series mentioned.

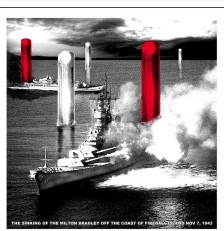

**Detailed Caption:** This image is a stylized artwork depicting a naval scene with a surreal twist. The central focus is a large battleship, rendered in grayscale, with its bow pointed towards the bottom right corner of the image. The ship is shown with multiple gun turrets and a superstructure crowded with details such as antennas, masts, and hull features. Surrounding the battleship are several cylindrical structures, some of which are tipped over, creating a chaotic and disordered appearance. These structures are colored in red and white, with the red parts appearing to have a reflective, metallic texture. The cylindrical shapes are of varying sizes and are dispersed throughout the image, with some partially submerged in the water. The background features a calm sea with a slight ripple texture, and the horizon is visible with a cloudy sky above. The sky is filled with dark clouds, suggesting an overcast or stormy weather condition. The image also contains text at the bottom, which reads "THE SINKING OF THE MILTON BRADLEY OFF THE COAST OF FIREBALL ISLAND NOV 7, 1943." This text provides a context to the image, indicating that the artwork is a stylized representation of an event involving the Milton Bradley company and a location known as Fireball Island, with a date of November 7, 1943. The overall composition of the image is dynamic, with the juxtaposition of the battleship and the cylindrical structures creating a sense of disruption and destruction. The use of grayscale for the ship and the red and white for the structures creates a stark contrast, drawing attention to the unusual elements in the scene.

Table 8: Visualizations on the image description in **DenseFusion-1M**. For better visualization, information about objects/attributes, spatial positions, text information, and knowledge/reasoning are marked in individual colors.

