# Appendix

## A  Motivation

**For what purpose was the dataset created?** Was there a specific task in mind? Was there a specific gap that needed to be filled? Please provide a description.

- This dataset was created to support advanced research on Multimodal Language Learning Models (MLLMs). It provides hyper-detailed descriptions of diverse visual cues, aiming to bridge the gap of the lack of hyper-detailed image-text pairs in the field.

**Who created the dataset (e.g. which team, research group) and on behalf of which entity (e.g. company, institution, organization)?**

- The original dataset is authorized by LAION AI (`https://laion.ai/`).
- This dataset annotation is presented by Beijing Academy of Artificial Intelligence, BAAI in short. Contributors include Xiaotong Li (Peking University), Fan Zhang (BAAI), Haiwen Diao (Dalian University of Technology), Yueze Wang (BAAI), Xinlong Wang (BAAI), Ling-Yu Duan (Peking University).

**Who funded the creation of the dataset?** If there is an associated grant, please provide the name of the grantor and the grant name and number.

- The dataset was funded by the Beijing Academy of Artificial Intelligence. The original data resources are publicly avaiable from LAION AI.

**Any other comments?**

- LAION dataset is a widely-used and publicly accessible data source. We follow the previous research and go deeper into excavating the hyper-detailed information inside LAION images. We believe this can inspire the MLLM community for comprehensive visual perception.

## B  Composition

**What do the instances that comprise the dataset represent (e.g. documents, photos, people, countries)?** Are there multiple types of instances (e.g., movies, users, and ratings; people and interactions between them; nodes and edges)? Please provide a description.

- The dataset primarily consists of images sourced from LAION-2B, which is a comprehensive dataset of diverse image types gathered from the internet. It contains a wide range of image categories including photos, visual art, commercial design, and infographics, etc.

**How many instances are there in total (of each type, if approriate)?**

- In total, the dataset **DenseFusion-1M** contains approximately 1,059K image-text pairs, while **DenseFusion-4V-100K** includes 100K image-text pairs. This provides a comprehensive overview of the number of instances for each dataset type.

**Does the dataset contain all possible instances or is it a sample (not necessarily random) of instances from a larger set?** If the dataset is a sample, then what is the larger set? Is the sample representative of the larger set (e.g., geographic coverage)? If so, please describe how this representativeness was validated/verified. If it is not representative of the larger set, please describe why not (e.g., to cover a more diverse range of instances, because instances were withheld or unavailable).

- The dataset for GPT-generated data includes all instances obtained, no sampling has been applied. For the data generated by our caption process. We manually filter out captions with incomplete or repetitive descriptions. The primary goal of this selection is to enhance the quality and diversity of the data rather than to provide a representative sample of all potential instances. No formal validation or verification of representativeness has been conducted due to the nature of the filtering process.

**What data does each instance consist of?** "Raw" data (e.g., unprocessed text or images)or features? In either case, please provide a description.

- Each instance consists of the image id, image url, and image description.

**Is there a label or target associated with each instance?** If so, please provide a description.

- The label corresponds to the image caption with detailed description on visual elements, including OCR, multiple objects, spatial relation, etc.

**Is any information missing from individual instances?** If so, please provide a description, explaining why this information is missing (e.g., because it was unavailable). This does not include intentionally removed information, but might include, e.g. redacted text.

- There is no missing information.

**Are relationships between individual instances made explicit (e.g. users' movie ratings, social network links)?** If so, please describe how these relationships are made explicit.

- There is no explicit relationship between individual instances.

**Are there recommended data splits (e.g., training, development/validation, testing)?** If so, please provide a description of these splits, explaining the rationale behind them.

- For the purposes of multimodal pre-training in this paper, we primarily use the image descriptions as our training dataset. Currently, there are no dedicated development/validation or testing sets.

**Are there any errors, sources of noise, or redundancies in the dataset?** If so, please provide a description.

- After our careful examination, the dataset is generally detailed and accurate. In particular, it inevitably contains data noise from the captioning results, which is a common issue in large-scale datasets.

**Is the dataset self-contained, or does it link to or otherwise rely on external resources (e.g. websites, tweets, other datasets)?** If it links to or relies on external resources, a) are there guarantees that they will exist, and remain constant, over time; b) are there official archival versions of the complete dataset (i.e., including the external resources as they existed at the time the dataset was created); c) are there any restrictions (e.g., licenses, fees) associated with any of the external resources that might apply to a dataset consumer? Please provide descriptions of all external resources and any restrictions associated with them, as well as links or other access points, as appropriate.

- The images are sourced from the public LAION dataset, which is widely used and well-maintained by LAION AI. It is legally licensed and includes images from external web links. Researchers can easily download the images using their corresponding URLs. Thanks to the diligent maintenance by LAION AI, the MLLM community has made significant progress in the fields of vision-language understanding and reasoning.

**Does the dataset contain data that might be considered confidential (e.g. data that is protected by legal privilege or by doctor-patient confidentially, data that includes the content of individuals' non-public communications)?** If so, please provide a description.

- No. After our careful examination, there is no data that might be considered confidential. The information from the dataset are generated by either GPT-4V or our public caption engine.

**Does the dataset contain data that, if viewed directly, might be offensive, insulting, threatening, or might otherwise cause anxiety?** If so, please describe why.

- The image resources are from public LAION dataset and have been filtered at certain extent.

**Does the dataset identify any subpopulations (e.g. by age, gender)?** If so, please describe how these subpopulations are identified and provide a description of their respective distributions within the dataset.

- No.

**Is it possible to identify individuals (i.e., one or more natural persons), either directly or indirectly (i.e., in combination with other data) from the dataset?** If so, please describe how.

- No, we emphasize all data are from publicly available LAION that has a good reputation in human privacy protecting. Even human images are included in this dataset, they are anonymous by LAION AI.

**Does the dataset contain data that might be considered sensitive in any way (e.g., data that reveals race or ethnic origins, sexual orientations, religious beliefs, political opinions or union memberships, or locations; financial or health data; biometric or genetic data; forms of government identification, such as social security numbers; criminal history)?** If so, please provide a description.

- No.

**Any other comments?**

- No.

## C   Collection Process

**How was the data associated with each instance acquired?** Was the data directly observable (e.g., raw text, movie ratings), reported by subjects (e.g., survey responses), or indirectly inferred/derived from other data (e.g., part-of-speech tags, model-based guesses for age or language)? If the data was reported by subjects or indirectly inferred/derived from other data, was the data validated/verified? If so, please describe how.

- The images are from public resources. The image descriptions are generated from caption engine based on open-sourced LLaVA.

**What mechanisms or procedures were used to collect data (e.g. hardware apparatuses or sensors, manual human curation, software programs, software APIS)?** How were these mechanisms or procedures validated?

- The dataset in constructed from either advanced GPT-4V or the multi-modality caption engine.

**If the dataset is a sample from a larger set, what was the sampling strategy (e.g. deterministic, probabilistic with specific sampling probabilities)?**

- The data processing strategy is displayed in the main paper 3.1 in detail.

**Who was involved in the data collection process (e.g. students, crowdworkers, contractors) and how were they compensated (e.g. how much were crowdworkers paid)?**

- Data collection process is fully conducted by authors.

**Over what timeframes was the data collected?** Does this timeframe match the creation timeframe of the data associated with the instances (e.g., recent crawl of old news articles)? If not, please describe the timeframe in which the data associated with the instances was created.

- Our dataset is collected in May 2024. No instance is associated with timeframe.

**Any other comments?**

- No.

## D Preprocessing/cleaning/labeling

**Was any preprocessing/cleaning/labeling of the data done (e.g., discretization or bucketing, tokenization, part-of-speech tagging, SIFT feature extraction, removal of instances, processing of missing values)?** If so, please provide a description. If not, you may skip the remaining questions in this section.

- The data processing strategy is displayed in the main paper 3.1 in detail.

**Was the "raw" data saved in addition to the preprocessed/cleaned/labeled data (e.g., to support unanticipated future uses)?** If so, please provide a link or other access point to the "raw" data.

- Yes, source dataset and original file names are saved.

**Is the software that was used to preprocess/clean/label the data available?** If so, please provide a link or other access point.

- Yes, GPT-API for instruction and template generation is available on `https://openai.com/chatgpt`.

**Any other comments?**

- No.

## E Uses

**Has the dataset been used for any tasks already?** If so, please provide a description.

- The dataset has been used for multi-modal understanding and reasoning tasks, producing consistent and signficant improvements.

**Is there a repository that links to any or all papers or systems that use the dataset?** If so, please provide a link or other access point.

- There is a repository that will include the above information.

**What (other) tasks could the dataset be used for?**

- The dataset could be used for contrastive vision-language pre-training and conditioned image generation.

**Any other comments?**

- No.

## F Distribution

**Will the dataset be distributed to third parties outside of the entity (e.g., company, institution, organization) on behalf of which the dataset was created?** If so, please provide a description.

- Yes, the dataset will be publicly available.

**How will the dataset will be distributed (e.g., tarball on website, API, GitHub)?** Does the dataset have a digital object identifier (DOI)?

- The dataset is distributed on Huggingface (`https://huggingface.co/datasets/BAAI/DenseFusion-1M`). No DOI for the dataset.
- The Croissant metadata can be accessed through official api `https://huggingface.co/api/datasets/BAAI/DenseFusion-1M/croissant` provided by HuggingFace.

**When will the dataset be distributed?**

- The dataset will released in July 2023.

**Will the dataset be distributed under a copyright or other intellectual property (IP) license, and/or under applicable terms of use (ToU)?** If so, please describe this license and/or ToU, and provide a link or other access point to, or otherwise reproduce, any relevant licensing terms or ToU, as well as any fees associated with these restrictions.

- The dataset annotation is publicly available under license of Creative Commons licenses (CC-BY). This project utilizes certain datasets and checkpoints that are subject to their respective original licenses. Users must comply with all terms and conditions of these original licenses. The content of this project itself is licensed under the Apache license 2.0.

**Any other comments?**

- No.

## G Maintenance

**Who will be supporting/hosting/maintaining the dataset?**

- The dataset is supporting/maintaining by Beijing Academy of Artificial Intelligence (BAAI).

**How can the owner/curator/manager of the dataset be contacted (e.g., email address)?**

- The authors can be contacted through their emails.

**Is there an erratum?** If so, please provide a link or other access point.

- No.

**Will the dataset be updated (e.g., to correct labeling errors, add new instances, delete instances)?** If so, please describe how often, by whom, and how updates will be communicated to dataset consumers (e.g., mailing list, GitHub)?

- Updates of the dataset will be posted at Huggingface `https://huggingface.co/datasets/BAAI/DenseFusion-1M`.

**If others want to extend/augment/build on/contribute to the dataset, is there a mechanism for them to do so?** If so, please provide a description. Will these contributions be validated/verified? If so, please describe how. If not, why not? Is there a process for communicating/distributing these contributions to dataset consumers? If so, please provide a description.

- Yes, the contribution will be considered and validated, through the huggingface community.

**Any other comments?**

- No.