# OpenReview forum: "DenseFusion-1M: Merging Vision Experts for Comprehensive Multimodal Perception"
_NeurIPS.cc/2024/Datasets_and_Benchmarks_Track — NeurIPS 2024 Track Datasets and Benchmarks Poster_

### Official Review · Reviewer_TaN1 · 2024-07-24

**Rating:** 7
**Confidence:** 3
**Correctness:** yes
**Clarity:** yes, good writing

**Review:**

Weaknesses:
1. Although the authors selected the images from the LAION dataset, LAION itself may have some biases, such as geographical, cultural, and social background. These biases could potentially be inherited into the DenseFusion-1M dataset, thus affecting the fairness and representativeness of the dataset. The article does not discuss in detail how these potential biases can be assessed and how measures can be taken to mitigate or eliminate the effects of these biases. This is critical to ensure that the dataset is fair and widely used.

2. Although the authors mention the use of a low-cost MLLM as the information fusion centre, the implementation of the entire Perceptual Fusion framework and the generation of large-scale datasets still requires significant computational resources. This includes parallel computation and storage requirements for multiple steps such as image processing, object detection, and text recognition. For larger datasets or more complex vision tasks, the computational cost may increase further, thus limiting the scalability and practical application of the framework.

3. The article does not explain sufficiently the details of how different vision experts in the Perceptual Fusion framework specifically contribute to the final image description. This makes it difficult for readers to gain a deeper understanding of how the framework works and its performance benefits.

4. No visual analyses or case studies are provided to demonstrate the specific roles and contributions of different visual experts in the process of generating image descriptions. This may weaken the reader's confidence in the effectiveness of the framework.

**Strengths:**

Strengths:
1. The Perceptual Fusion framework proposed in this article provides an innovative approach for generating highly detailed image descriptions by integrating the capabilities of multiple vision experts. This approach transcends the traditional limitations of a single model or a single dataset, and significantly improves the accuracy and detail of image descriptions by combining the strengths of multiple experts.

2. The authors carefully selected 1 million images from the LAION dataset and generated the DenseFusion-1M dataset using the Perceptual Fusion framework. This dataset is not only large in quantity but also high in quality, containing rich textual information, multiple objects, attributes, spatial relationships, and external knowledge. These high-quality image descriptions provide strong support for training and enhancing the comprehensive visual perception ability of MLLMs.

3. The article validates the effectiveness of the DenseFusion-1M dataset on several visual-linguistic benchmark tests through extensive experiments. These benchmark tests cover different visual and linguistic tasks and fully demonstrate the broad applicability of the dataset in enhancing the perceptual and cognitive abilities of MLLMs. These advantages not only enhance the performance of the model, but also provide new directions and data support for future research.

**Additional Feedback:**

N.a

**Documentation:**

yes

**Ethics:**

N.A

**Limitations:**

see above

**Opportunities For Improvement:**

Questions:
1. How do you assess and minimize possible biases in the DenseFusion-1M dataset, especially those inherited from the LAION dataset?

2. How much computational resources are required to generate a large dataset like DenseFusion-1M in a real application? Are there feasible ways to reduce the computational cost?

3. In the Perceptual Fusion framework, how is the information from different vision experts integrated into the final image description? Are there specific fusion algorithms or mechanisms?

4. Can your framework be easily extended to other types of vision tasks or more complex vision scenarios?

5. What measures are in place to handle the noise in data gathered by visual experts?

6. How does the dataset ensure the accuracy and reliability of the hyper-detailed descriptions?

**Relation To Prior Work:**

yes, it clearly discussed the relation to prior works.

**Summary And Contributions:**

This article presents the DenseFusion-1M dataset, which aims to generate high-quality, highly detailed image-text pairs by fusing the capabilities of multiple visual experts to support the research and application of Multimodal Large Language Models (MLLMs) in integrated visual perception. This approach aims to overcome the limitations of current caption engines and improve the perception and cognition abilities of MLLMs. Experiments show that this dataset significantly improves the perceptual and cognitive abilities of existing MLLMs in several visual-verbal benchmark tests.

---

> ### Author Rebuttal · Authors · 2024-08-17
>
> Thanks for your positive and valuable suggestions. The responses to your main concerns are listed below.
>
> **Q1**: `Potential biases inherited from LAION. How to minimize it in DenseFusion-1M?`
>
> **A1**: We fully agree that it is crucial to explore. Some efforts are as follows:
>
> (1) The images from LAION has been processed by employing automated filtering techniques, e.g. CLIP embeddings and specialized classifiers, for filtering harmful or inappropriate contents.
>
> (2) We try to alleviate potential biases by Semantic Clustering strategy during data pre-processing (in Section 3.1). It involves balanced sampling based on image semantics and ensures image diversity, mitigating biases from an uneven image distribution.
>
> (3) We plan to enhance our approach with more strategies:
> - **[Diverse source construction.]** For more data diversity, we will extend our dataset by integrating images from multi-sources, e.g. DataComp[1] and COYO[2]. This helps in balancing the dataset and minimizing inherent biases.
> - **[Post-processing approaches.]** To ensure the fairness, we will develop a Large Language Model (LLM) that further justifies the image and caption contents, systematically eliminating any observed biases.
>
>
> [1] DataComp-1B: In search of the next generation of multimodal datasets. NeurIPS 2023.
>
> [2] COYO-700M: Image-text pair dataset. 2023.
>
> **Q2**: `(a) Computational resources for DenseFusion-1M? (b) Feasible ways to reduce such a cost?`
>
> **A2**: Here are detailed computational costs for DenseFusion-1M.
>
>
> |DenseFusion-1M	 | Visual Experts Preparations | Perceptual Fusion Generation |
> |---|---|---|
> |Computational resources | 16 A100 GPUs | 16 A100 GPUs |
> |Time cost | 4 hours | 10 hours |
>
> (a) The whole pipeline is meticulously designed for both cost-effectiveness and efficiency, enabling to generate 1 million data points in one single day via 8 A100 GPUs. It allows to facilitate the scalable high-quality datasets in a low-budget manner.
>
> (b) From the above Table, the caption engine requires the most resources due to the large capacity of its component LLM. Some strategies are feasible for further benefits:
>
> - **[Efficient data engine design.]** Our framework is flexible and can allow other smaller yet powerful LLMs (e.g., Phi-3 3B [1], Qwen-2 3B [2]) to further improve efficiency.
>
> - **[Fast development strategies.]** Tools like SGLang [3] can be used to accelerate the model inference, especially for large-scale dataset generation.
>
> [1] Phi-3 Technical Report. Arxiv 2024.
>
> [2] Qwen2 Technical Report. Arxiv 2024.
>
> [3] SGLang: Efficient Execution of Structured Language Model Programs. Arxiv 2023.
>
>
>
> **Q3**: `Extension to other more complex vision scenarios? Parallel computation and storage issues for larger datasets.`
>
> **A3**: Our framework is modular and flexible. Users can easily extend it to other tasks by balancing the available resources and involving more appropriate visual experts for enhancing MLLMs' capability.
>
> - **[Computation resources.]** Our pipeline is highly-efficient for large-scale generation (generating 1 million data in one day with 8 A100 GPUS).
>
> - **[Storage overhead.]** Various experts' outputs in DenseFusion-1M are stored in the text-json format. Its cost is significantly low (~ 1.6 GB) v.s. image storage (~ 744 GB), making it affordable even with more vision experts.
>
> **Q4**: `(a) How different vision expert information be integrated? (b) Are there specific algorithms or mechanisms?`
>
> **A4**: (a) The information from various experts serves as contextual inputs to enhance our engine's perceptual capabilities. We meticulously design the prompt template as an explicit perceptual fusion instruction (as detailed in Supply. C.2), encouraging the engine to fully leverage these image priors and enrich the final descriptions.
>
> (b) Besides, its fusion strategy learns from DenseFusion-4V-100K (generated from GPT-4V), which implicitly guides the engine to learn how to optimally integrate contextual information from different vision experts.
>
> **Q5**: `How different vision experts specifically contribute to the final image description. Visual analyses or case studies to show their specific roles and contributions.`
>
> **A5**: Our framework leverages various expertise for enhancing MLLMs' perceptual abilities. We display their specific contributions in the Figure 1-3 of [rebuttal.PDF]. Note that involving vision experts can enhance perceptual descriptions with corresponding aspects, e.g., the OCR model aids the MLLM in recognizing text characters, including multilingual languages, while the detection model helps accurately identify objects at various granularities. These validate the distinct roles of vision experts for comprehensive and accurate final descriptions.
>
> **Q6**: `(a) Measurement to handle visual experts noise? (b) How to ensure the accuracy and reliability?`
>
> **A6**: (a) Pre-processing on visual perceptions.
>
> - **[Visual Expert Choice:]** We prioritize advanced visual experts in specific domains to ensure the high-quality and reliable predictions.
>
> - **[High-confidence threshold:]** In Section 3.2.1, we implement a high confidence threshold for visual experts' predictions to effectively handle noise.
>
> - **[Self-refinement:]** We carefully designed prompt template for GPT-4V to refine inaccuracies from visual experts based on real image contents, thereby implicitly encouraging our engine to mitigate noise.
>
> (b) Quality for hyper-detailed descriptions.
>
> - In the early stage, we iteratively polish our designed pipeline, ensuring an optimal one for high-quality dataset construction.
> - After that, we check the whole dataset and eliminate low-quality captions through heuristic rules, e.g. repetitive and incomplete issues, etc.
> - We plan to develop a Large Language Model (LLM) to justify the descriptions and systematically filter more bias types.
>
> Thank you once again for your time and consideration. We anticipate your feedback and remain hopeful for a positive evaluation.

---

### Official Review · Reviewer_r1id · 2024-07-24
**Interesting paper but needs some clarifications**

**Rating:** 6
**Confidence:** 3
**Correctness:** See above.
**Clarity:** See above.

**Review:**

The paper needs a bit of rewriting, as there are certain passages that are not clear to me. The main point that I fail to understand is how much faster and cheaper is to annotate the dataset with the proposed pipeline with respect to using GPT-4V. I don't think there is any study on the improvement done in this respect by the pipeline proposed by the authors.

Other (less important) points that I am missing are:

1. The authors use often the term OCR without ever explicitly saying what it means.
2. The authors do not report what is the metric recorded in Table 2.
3. At page 4 the authors write that they select the top 20 images, what does top mean in this setting.
4. I am not sure I understand the results in Table 4. Did you only pre-train with your dataset the model labelled with (Ours)? To have more conclusive results wouldn't have been better to pre-train different models and see that the models not pre-trained perform worse than the ones pre-trained?
5. I have the same problem with Table 3 and 4 as for Table 2, which metric are you measuring?
6. What does it mean SEED(%) and TextVQA(%) in Figure 5? I thought those were names of tasks, not metrics.

Additional comment:

1. The images are often blurred and difficult to read

**Strengths:**

The paper proposes an interesting dataset which helps in pre-training MLLMs.

**Additional Feedback:**

No additional feedback.

**Documentation:**

Yes there is enough documentation.

**Ethics:**

No.

**Limitations:**

Yes they have been discussed.

**Opportunities For Improvement:**

See comments above.

**Relation To Prior Work:**

See above.

**Summary And Contributions:**

The paper proposes a new annotating pipeline called "perceptual fusion" to produce high quality captions (with a lot of details) at a very low cost. The authors showcase the effectiveness of the proposed pipeline by generating the dataset DenseFusion-1M (indeed containing 1M images with detailed captions) and then by using it to pre-train the models.

---

> ### Author Rebuttal · Authors · 2024-08-17
>
> Thanks for your approval and valuable suggestions. We will carefully revise the manuscript according to your feedback and improve clarity.
>
> **Q1:**` Study the time and cost comparisons for annotating the dataset using the proposed pipeline compared to using GPT-4V.`
>
> **A1**: Thanks for your valuable suggestion. We conduct a comprehensive comparison between our proposed DenseFusion pipeline and the GPT-4V approach for dataset annotation. In summary, DenseFusion offers a highly cost-effective strategy for scaling up the volume of high-quality data, **significantly reducing approximately 20x time and 50x economic costs compared to GPT-4V approach.**
>
> |Comparison	| DenseFusion | GPT4V |
> |---|---|---|
> |Resulted Dataset | DenseFusion-1M | DenseFusion-4V-100K |
> |Device	| Local 16 A100 GPUs | Remote API server |
> |Time | 14 hours (1 million) | 30 hours (0.1 million) |
> |Cost | ~200 Dollars (1 million) | ~1,000 Dollars (0.1 million) |
>
> As demonstrated in the above Table, DenseFusion operates efficiently on local GPU devices, eliminating the need for costly remote API servers and achieving a substantial reduction in both time and cost. Its impressive performance and low-budget consumption pave the way for a new direction in the MLLM community, enabling the scalable production of high-quality and detailed descriptions.
>
> **Q2**: `Explicitly clarify what the term OCR means.`
>
> **A2**: Throughout this paper, OCR is the abbreviation of Optical Character Recognition (OCR), which refers to the text recognition model we employed to extract visible text contents in the images. We will revise the manuscript to explicitly clarify this term upon its first mention, ensuring clarity for all readers.
>
> **Q3**: `What is the metric recorded in Table 2, Table 3 and 4. Which metric are you measuring? `
>
> **A3**: The metrics presented in Tables 2, 3, and 4 represent the accuracy percentage (%) of correct answers across all questions. The evaluation follows the same standard protocols as those used in prior works (e.g., LLaVA-1.5[1] and ShareGPT4V[2]) on 10 widely used MLLM benchmarks, including Visual Question Answering (VQA) tasks and multi-modality understanding tasks.
>
> [1] Visual Instruction Tuning (LLaVA-1.5). NeurIPS 2023.
>
> [2] ShareGPT4V: Improving Large Multi-Modal Models with Better Captions. ECCV 2024.
>
> **Q4**: `At page 4 the authors write that they select the top 20 images, what does top mean in this setting.`
>
> **A4**: In this context, "top" refers to the images that are closest to the cluster centers in the clustering process. Following the procedure of SemDeDup (illustrated in Fig. 1 of SemDeDup [1]), we select the 20 images closest to each cluster center from the deduplicated subset. Thank you for your suggestion, we will make this clearer in the revised manuscript.
>
>
>
> **Q5**: `What does it mean SEED(%) and TextVQA(%) in Figure 5? I thought those were names of tasks, not metrics.`
>
> **A5**: The notation is short for accuracy percentage (%) in these benchmarks for indicating the metric in the figure. Thanks for your reminder, we will clarify this in the revised manuscript.
>
> **Q6**: `(a) I am not sure I understand the results in Table 4. Did you only pre-train with your dataset the model labelled with (Ours)? (b) To have more conclusive results wouldn't have been better to pre-train different models and see that the models not pre-trained perform worse than the ones pre-trained?`
>
> **A6**:
> Thanks for the questions. (a) It is not certain whether this question refers to Table 4 or Table 2, so we will response in both aspects.
>
> - Table 4: The ablation study in Table 4 examines the impact of visual encoders under Baseline, Frozen, and Tuning settings. The Baseline model, LLaVA-1.5, is **not pre-trained** with our dataset, while the Frozen and Tuning models are **pre-trained**. Table 4 highlights the benefits of our dataset across various training settings for the visual encoder.
>
> - Table 2: In Table 2, the model labelled with (Ours) denotes that model is **pre-trained** on our DenseFusion-1M dataset, while all other settings remained identical to the model not labeled as (Ours). The experiment in Table 2 of main paper demonstrates the significant and consistent improvements in various benchmarks provided by our DenseFusion-1M dataset.
>
> (b) In main paper, we conducted experiments on different model designs, such as LLaVA-1.5[1] and S2[2], as shown in Table 2. To further address your concerns, we conduct more experiments with different language models, as shown in **Table 1 in [rebuttal.PDF]**, including Vicuna[3], LLaMA3[4], and Qwen2[5]. These results comprehensively demonstrate our solid and consistent improvements, verifying our effectiveness across various models.
>
>
> [1] Visual Instruction Tuning (LLaVA-1.5). NeurIP 2023.
>
> [2] When Do We Not Need Larger Vision Models? (S2). ECCV 2024.
>
> [3] Vicuna: An Open-Source Chatbot Impressing GPT-4 with 90% ChatGPT Quality.
>
> [4] The Llama 3 Herd of Models. Arxiv 2024.
>
> [5] Qwen2 Technical Report. Arxiv 2024.
>
>
> | Model| SQA_I | VQA_v2 | GQA | VQA_T| MME | MMB | SEED_I| POPE | MMVet|
> |---|---|---|---|---|---|---|---|---|---|
> | LLaVA-1.5 (Vicuna-7B) | 66.8 | 78.5 | 62.0 | 58.2 | 1510| 64.3 | 66.2 | 85.9 | 30.5 |
> | DenseFusion (Vicuna-7B) | 69.3 | 80.8 | 64.0 | 62.0 | 1574 | 69.2 | 70.1 | 86.5 | 37.8 |
> | LLaVA-1.5 (LLaMA3-8B) | 72.3 | 79.7 | 63.8 | 58.7 | 1553 | 72.8 | 69.2 | 85.0 | 34.9 |
> | DenseFusion (LLaMA3-8B) | 72.9 | 80.4 | 64.4 | 61.0 | 1560 | 73.4 | 71.6 | 85.3 | 40.0 |
> | LLaVA-1.5 (Qwen2-7B) | 72.3 | 79.8 | 63.4 | 57.0 | 1566 | 72.9 | 70.0 | 85.7 | 35.8 |
> | DenseFusion (Qwen2-7B) | 73.5 | 80.5 | 64.0 | 58.9 | 1528 | 73.5 | 71.6| 86.0 | 41.4|
>
>
> **Q7**: `The images are often blurred and difficult to read.`
>
> **A7**: Thank you for your careful review. We'll double-check all figures and ensure they are high-definition for clear readability.
>
> Thanks again for your time and consideration. We look forward to your feedback and remain hopeful for a favorable consideration.

---

> > ### Comment · Reviewer_r1id · 2024-08-30
> > **Raised my score**
> >
> > The authors have answered my questions, I raise my score trusting the authors to include these new results in the final paper. Additionally, I hope they will improve the writing as stated in their rebuttal.

---

> > ### Author Response · Authors · 2024-08-31
> >
> > We appreciate the reviewer’s feedback and are pleased to hear that our response can address your concerns. We will incorporate your valuable suggestions into the revised manuscript.
> >
> > Thank you once again for your insightful comments.
> >
> > Best regards,

---

> ### Author Response · Authors · 2024-08-30
>
> Dear Reviewer,
>
> We express our sincere gratitude to the reviewer for dedicating time to review our paper. We have meticulously provided comprehensive responses to address each of the concerns raised, including (1) the main question about time and cost comparisons in A1, and (2) other points about clarification in A2-A7.
>
> As the discussion deadline approaches, we look forward to your participation in the Reviewer-Author discussion phase, as your insights are invaluable for improving the quality of our paper. If you have any additional questions or require further clarification, please do not hesitate to inform us.
>
> If our responses have satisfactorily addressed your concerns, we kindly request a reconsideration of the score. We are more than willing to address any remaining concerns and ensure a comprehensive resolution. Thank you again for your time and consideration.
>
> Best regards,

---

### Official Review · Reviewer_BJnX · 2024-07-24

**Rating:** 8
**Confidence:** 3
**Correctness:** Yes
**Clarity:** Yes

**Review:**

The authors have developed an inexpensive data engine that focuses on providing more vision detail, enabled by fusing vision experts. Given the significant effort invested in crafting and making this high-quality dataset available to the public and the high potential benefits to MLLM development, I strongly suggest its acceptance.

**Strengths:**

1. The generation method is clearly described and inexpensive to execute, making it possible for others to build on this work.
2. The use of deduplication ensures diversity and variation by maximizing the diversity of image distribution using SemDeDup.
3. Annotation strategies are carefully designed using expert models.
4. Using a captioning model to mimic GPT-4V helps scale up to DenseFusion-1M by 10x while reducing query costs.
5. Experimental results using LLaVA-1.5 and LLaVA-S^2, which use DenseFusion-1M in pretraining, achieve consistently better results on common vision-language evaluation benchmarks.
6. Ablation studies show that the data efficiency is better than ShareGPT-4V under an equivalent sample scale.

**Additional Feedback:**

None.

**Documentation:**

Yes

**Limitations:**

1. The quality of this dataset highly depends on vision experts (AI models). While it seems adequate for MLLM research and bridging the gap between MLLM and vision experts, it is still limited by the capabilities of these vision experts.
2. The discussion on how human-in-the-loop can potentially be integrated into the engine to further enhance quality could be expanded.

**Opportunities For Improvement:**

1. Clarify whether the OCR model iteratively works over the detected bounding boxes. How the fusion model connects text and objects is not clear.
2. The usefulness of the world knowledge model is not justified well. Does it provide extra useful information (beyond tagging and GPT-4V, for example)? Please provide evidence supporting its usage.
3. Provide comparative experiments to demonstrate the quality of DenseFusion-1M compared to DenseFusion-4V-100K.

**Relation To Prior Work:**

Yes

**Summary And Contributions:**

This work proposes Perceptual Fusion, an inexpensive strategy that substantially enriches the current image caption dataset by fusing diverse vision experts. This approach effectively scales up the volume of high-quality data. The resulting DenseFusion-1M dataset is demonstrated to be beneficial in MLLM pretraining, achieving significant improvements across benchmarks.

---

> ### Author Rebuttal · Authors · 2024-08-17
>
> Thanks for the valuable and helpful suggestions for further improvement! The responses to your main concerns are listed below.
>
> **Q1**: `(a) Does the OCR model iteratively work on the detected bounding boxes? (b) How does the fusion model connect text and objects?`
>
> **A1**: (a) Our OCR model does not operate iteratively over the bounding boxes from our detection model. Instead, we have decoupled the processes of text recognition and object recognition to obtain accurate predictions via two specialized experts respectively. The adopted OCR model is specialized in recognizing visible text contents, while open/closed-set detection models are employed to effectively recognize common objects.
>
> (b) After independently extracting perceptual information through these specialized models, the fusion model integrates them as contextual information by employing the carefully designed prompt template (as detailed in Supplementary C.2), which does not require an explicit connection.
>
>
> **Q2**: `Please provide evidence supporting its usage of the world knowledge model. Does it provide extra useful information (beyond tagging and GPT-4V, for example)? `
>
> **A2**: (a) As highlighted in prior works [1,2], world knowledge could provide extra useful information that are often not present within the visual content of images, e.g., the specific usage of products, background details, the identification of celebrities, etc. These useful information is valuable for addressing knowledge-based and real-world problems, but is difficult to extract solely through visual perception models.
>
> (b) Visualization examples: To further illustrate our point, we refer to the visualizations in Figure 1 of the [rebuttal PDF], which demonstrate how world knowledge enriches the descriptions. For instance, it can supply the product type for the headphones and provide the geographical background for the map. Therefore, our introduced knowledge serves as a key role to enhance the knowledge depth and information density of the descriptions.
>
>
> [1] CapsFusion: Rethinking Image-Text Data at Scale. CVPR 2024.
>
> [2] VeCLIP: Improving CLIP Training via Visual-enriched Captions. ECCV 2024.
>
>
> **Q3**: `Provide comparative experiments to demonstrate the quality of DenseFusion-1M compared to DenseFusion-4V-100K.`
>
>
> **A3**: Following your suggestion, we conduct the requested comparative experiments with models pre-trained with 100K descriptions generated by DenseFusion and GPT-4V respectively.
>
> | Model | MMB | SEED | VQA_T | SQA_I |
> |---|---|---|---|---|
> | Baseline (LLaVA-1.5) | 64.3 | 58.6 | 58.2 | 66.8
> | DenseFusion-1M (100K)| 67.0 | 60.8 | 60.6 | 68.9 |
> | DenseFusion-4V-100K (100K) | 67.3 | 61.3 | 60.8 | 69.1 |
>
> The results demonstrate that our DenseFusion approach achieves comparable performance to GPT-4V, while offering a substantial reduction in computational costs, **significantly reducing approximately 20x time and 50x economic costs compared to GPT-4V approach.**
>
> |Comparison	| DenseFusion | GPT4V |
> |---|---|---|
> |Resulted Dataset | DenseFusion-1M | DenseFusion-4V-100K |
> |Device	| Local 16 A100 GPUs | Remote API server |
> |Time | 14 hours (1 million) | 30 hours (0.1 million) |
> |Cost | ~200 Dollars (1 million) | ~1,000 Dollars (0.1 million) |
>
> As demonstrated, DenseFusion operates efficiently on local GPU devices, eliminating the need for costly remote API servers and achieving a substantial reduction in both time and cost. Its impressive performance and cost-effectiveness pave the way for a new direction in the MLLM community, enabling the scalable production of high-quality and detailed descriptions at an affordable cost.
>
>
>
> **Q4**: `The quality of this dataset highly depends on vision experts (AI models). While it seems adequate for MLLM research and bridging the gap between MLLM and vision experts, it is still limited by the capabilities of these vision experts. `
>
>
> **A4**: (1) We acknowledge the limitations of relying on vision experts, as their performance is inherently related to their current capabilities. While it is crucial to note that these specialized vision experts exhibit superior perceptual abilities compared to general MLLMs and are highly effective in providing detailed perceptual information to bridge the gaps.
>
> (2) Most importantly, as these models are still rapidly developing, we highlight that our flexible framework can continuously evolve alongside the enhanced capabilities of state-of-the-art vision experts.
>
>
> **Q5**: `The discussion on how human-in-the-loop can potentially be integrated into the engine to further enhance quality could be expanded.`
>
> **A5**: We sincerely appreciate your valuable advice on further improving the dataset quality. Introducing human-in-the-loop strategy can further improve quality of the generated dataset. Due to the labor-intensive nature of this approach, we plan to explore several practical implementations in our future work:
>
> - **[Quality Control.]** Implementing a quality control stage where human experts review a subset of the generated data can help maintain high standards.
>
> - **[Human Feedback.]** Human annotators can refine and validate the initial annotations provided by the caption engine. By reviewing and adjusting these annotations, we can develop a MLLM for learning to refine annotations and enhancing data quality.
>
> Thank you once again for your time and consideration. We look forward to your feedback and remain hopeful for a favorable consideration.

---

### Author Rebuttal · Authors · 2024-08-17

We sincerely thank all the reviewers for their valuable time in reviewing our manuscript. We are strongly encouraged by their positive reviews and constructive suggestions.

We thank all reviewers for their constructive comments. We are encouraged by your recognition of an innovative and interesting idea (Reviewer r1id, TaN1), an inexpensive and low-cost strategy paired with large high-quality datasets (Reviewer BJnX, r1id, TaN1), significant and consistent improvements across VLM benchmarks (Reviewer BJnX, TaN1), potential benefits and new directions for future research (Reviewer BJnX, TaN1). We believe our work significantly advances the boundaries of hyper-detailed image descriptions and provides a practical approach for scaling up high-quality synthetic image-text data, paving the way for the next generation of MLLM pre-training research.

All the answers (**A**) to questions (**Q**) are listed below. We conducted more experiments and analyses to address the reviewers’ concerns. The major supplements of figures and tables are provided in the [rebuttal.PDF]:

The major supplements are summarized as follows:

- Visual analysis on specific contribution of World Knowledge for final description in Figure 1.

- Visual analysis on specific contribution of OCR model for final description in Figure 2.

- Visual analysis on specific contribution of Detection Model for final description in Figure 3.

- Extended experiments with various methods and models pre-trained on DenseFusion-1M dataset in Table 1.


We hope that our responses have effectively addressed all concerns, and we would appreciate any further feedback you can provide.

Sincerely yours,

Authors.

---

### Author Response · Authors · 2024-08-26

Dear Reviewers,

We sincerely thank you for your time and valuable comments. We have provided detailed responses and results, which we believe address your concerns. We would appreciate the opportunity to discuss whether your concerns have been fully resolved, please let us know if you still have any unclear parts of our work.

Thank you once again for your time and consideration. We look forward to your feedback and remain hopeful for a favorable consideration.

---

### Comment · Area_Chair_NXMY · 2024-08-30
**Urgent: Reminder to Review Author Rebuttals and Engage in Discussion**

Dear Reviewers,

We are just two days away from the end of the discussion period. The authors have provided their rebuttals to the reviews. It is very important that you review the rebuttal and other reviews, and engage in the conversation or respond if the rebuttal addresses your concerns.

Thank you.

---

### Decision · Program_Chairs · 2024-09-26

**Decision:**

Accept (Poster)

**Comment:**

This paper proposes a low-cost synthetic captioning pipeline for generating a dataset with descriptive captions. The approach integrates rich information from perception models, such as image tagging, object detection, and OCR, using an inexpensive fusion VLM to produce dense and descriptive captions. The resulting dataset, DenseFusion-1M, is demonstrated to enhance several state-of-the-art VLMs across multiple benchmarks.

All three reviewers recommended accepting the paper. They i) found the curation of high-quality and diverse images appropriate, ii) considered the descriptive caption synthesis pipeline innovative and cost-effective, and iii) appreciated the experiments showing that the new dataset significantly improves VLMs on various metrics and benchmarks, as well as the ablations that justify the design choices. During the rebuttal, the authors provided additional experiments demonstrating that the improvements are consistent across different models, the effectiveness of the pipeline compared to expensive commercial alternatives, and analyses addressing questions about different perceptual components. The AC agrees with the reviewers that both the introduced dataset and the data generation method offer significant benefits to the community and therefore recommends accepting the paper. The authors should incorporate these new results into the final paper.